# An Information-Theoretic Perspective on Variance-Invariance-Covariance Regularization

**Ravid Shwartz-Ziv**[*]
New York University

**Randall Balestriero**
Meta AI, FAIR

**Kenji Kawaguchi**
National University of Singapore

**Tim G. J. Rudner**
New York University

**Yann LeCun**
New York University & Meta AI, FAIR

## Abstract

Variance-Invariance-Covariance Regularization (VICReg) is a self-supervised learning (SSL) method that has shown promising results on a variety of tasks. However, the fundamental mechanisms underlying VICReg remain unexplored. In this paper, we present an information-theoretic perspective on the VICReg objective. We begin by deriving information-theoretic quantities for deterministic networks as an alternative to unrealistic stochastic network assumptions. We then relate the optimization of the VICReg objective to mutual information optimization, highlighting underlying assumptions and facilitating a constructive comparison with other SSL algorithms and derive a generalization bound for VICReg, revealing its inherent advantages for downstream tasks. Building on these results, we introduce a family of SSL methods derived from information-theoretic principles that outperform existing SSL techniques.

## 1 Introduction

Self-supervised learning (SSL) is a promising approach to extracting meaningful representations by optimizing a surrogate objective between inputs and self-generated signals. For example, Variance-Invariance-Covariance Regularization (VICReg) [7], a widely-used SSL algorithm employing a de-correlation mechanism, circumvents learning trivial solutions by applying variance and covariance regularization.

Once the surrogate objective is optimized, the pre-trained model can be used as a feature extractor for a variety of downstream supervised tasks such as image classification, object detection, instance segmentation, or pose estimation [15, 16, 48, 67]. Despite the promising results demonstrated by SSL methods, the theoretical underpinnings explaining their efficacy continue to be the subject of investigation [5, 42].

Information theory has proved a useful tool for improving our understanding of deep neural networks (DNNs), having a significant impact on both applications in representation learning [3] and theoretical explorations [60, 72]. However, applications of information-theoretic principles to SSL have made unrealistic assumptions, making many existing information-theoretic approaches to SSL of limited use. One such assumption is to assume that the DNN to be optimized is stochastic—an assumption that is violated for the vast majority DNNs used in practice. For a comprehensive review on this topic, refer to the work by Shwartz-Ziv and LeCun [62].

---

[*]Correspondence to: `ravid.shwartz.ziv@nyu.edu`.

37th Conference on Neural Information Processing Systems (NeurIPS 2023).

In this paper, we examine Variance-Invariance-Covariance Regularization (VICReg), an SSL method developed for deterministic DNNs, from an information-theoretic perspective. We propose an approach that addresses the challenge of mutual information estimation in deterministic networks by transitioning the randomness from the networks to the input data—a more plausible assumption. This shift allows us to apply an information-theoretic analysis to deterministic networks. To establish a connection between the VICReg objective and information maximization, we identify and empirically validate the necessary assumptions. Building on this analysis, we describe differences between different SSL algorithms from an information-theoretic perspective and propose a new family of plug-in methods for SSL. This new family of methods leverages existing information estimators and achieves state-of-the-art predictive performance across several benchmarking tasks. Finally, we derive a generalization bound that links information optimization and the VICReg objective to downstream task performance, underscoring the advantages of VICReg.

Our key contributions are summarized as follows:

1. We introduce a novel approach for studying deterministic deep neural networks from an information-theoretic perspective by shifting the stochasticity from the networks to the inputs using the Data Distribution Hypothesis (Section 3)
2. We establish a connection between the VICReg objective and information-theoretic optimization, using this relationship to elucidate the underlying assumptions of the objective and compare it to other SSL methods (Section 4).
3. We propose a family of information-theoretic SSL methods, grounded in our analysis, that achieve state-of-the-art performance (Section 5).
4. We derive a generalization bound that directly links VICReg to downstream task generalization, further emphasizing its practical advantages over other SSL methods (Section 6).

## 2    Background & Preliminaries

We first introduce the necessary technical background for our analysis.

### 2.1    Continuous Piecewise Affine (CPA) Mappings.

A rich class of functions emerges from piecewise polynomials: spline operators. In short, given a partition $\Omega$ of a domain $\mathbb{R}^D$, a spline of order $k$ is a mapping defined by a polynomial of order $k$ on each region $\omega \in \Omega$ with continuity constraints on the entire domain for the derivatives of order $0,\ldots,k-1$. As we will focus on affine splines ($k = 1$), we only define this case for clarity. A $K$-dimensional affine spline $f$ produces its output via $f(\boldsymbol{z}) = \sum_{\omega \in \Omega}(\boldsymbol{A}_\omega \boldsymbol{z} + \boldsymbol{b}_\omega)\mathbb{1}_{\{\boldsymbol{z} \in \omega\}}$, with input $\boldsymbol{z} \in \mathbb{R}^D$ and $\boldsymbol{A}_\omega \in \mathbb{R}^{K \times D}, \boldsymbol{b}_\omega \in \mathbb{R}^K, \forall \omega \in \Omega$ the per-region *slope* and *offset* parameters respectively, with the key constraint that the entire mapping is continuous over the domain $f \in \mathcal{C}^0(\mathbb{R}^D)$.

**Deep Neural Networks as CPA Mappings.**    A deep neural network (DNN) is a (non-linear) operator $f_\Theta$ with parameters $\Theta$ that map a *input* $\boldsymbol{x} \in \mathbb{R}^D$ to a *prediction* $\boldsymbol{y} \in \mathbb{R}^K$. The precise definitions of DNN operators can be found in Goodfellow et al. [27]. To avoid cluttering notation, we will omit $\Theta$ unless needed for clarity. For our analysis, we only assume that the non-linearities in the DNN are CPA mappings—as is the case with (leaky-) ReLU, absolute value, and max-pooling operators. The entire input-output mapping then becomes a CPA spline with an implicit partition $\Omega$, the function of the weights and architecture of the network [51, 6]. For smooth nonlinearities, our results hold using a first-order Taylor approximation argument.

### 2.2    Self-Supervised Learning.

SSL is a set of techniques that learn representation functions from unlabeled data, which can then be adapted to various downstream tasks. While supervised learning relies on labeled data, SSL formulates a proxy objective using self-generated signals. The challenge in SSL is to learn useful representations without labels. It aims to avoid trivial solutions where the model maps all inputs to a constant output [11, 36]. To address this, SSL utilizes several strategies. *Contrastive methods* like SimCLR and its InfoNCE criterion learn representations by distinguishing positive and negative examples [16, 54]. In contrast, *non-contrastive* methods apply regularization techniques to prevent the collapse [14, 17, 28].

**Variance-Invariance-Covariance Regularization (VICReg).** A widely used SSL method for training joint embedding architectures [7]. Its loss objective is composed of three terms: the invariance loss, the variance loss, and the covariance loss:

- *Invariance loss:* The invariance loss is given by the mean-squared Euclidean distance between pairs of embedding and ensures consistency between the representation of the original and augmented inputs.

- *Regularization:* The regularization term consists of two loss terms: the **variance loss**—a hinge loss to maintain the standard deviation (over a batch) of each variable of the embedding—and the **covariance loss**, which penalizes off-diagonal coefficients of the covariance matrix of the embeddings to foster decorrelation among features.

VICReg generates two batches of embeddings, $\boldsymbol{Z} = [f(\boldsymbol{x}_1), \dots, f(\boldsymbol{x}_B)]$ and $\boldsymbol{Z}' = [f(\boldsymbol{x}'_1), \dots, f(\boldsymbol{x}'_B)]$, each of size $(B \times K)$. Here, $\boldsymbol{x}_i$ and $\boldsymbol{x}'_i$ are two distinct random augmentations of a sample $I_i$. The covariance matrix $\boldsymbol{C} \in \mathbb{R}^{K \times K}$ is obtained from $[\boldsymbol{Z}, \boldsymbol{Z}']$. The VICReg loss can thus be expressed as follows:

$$\mathcal{L} = \underbrace{\frac{1}{K} \sum_{k=1}^{K} \left( \alpha \max\left(0, \gamma - \sqrt{\boldsymbol{C}_{k,k} + \epsilon}\right) + \beta \sum_{k' \neq k} (\boldsymbol{C}_{k,k'})^2 \right)}_{\text{Regularization}} + \underbrace{\eta \|\boldsymbol{Z} - \boldsymbol{Z}'\|_F^2 / N}_{\text{Invariance}}. \tag{1}$$

### 2.3 Deep Neural Networks and Information Theory

Recently, information-theoretic methods have played an essential role in advancing deep learning by developing and applying information-theoretic estimators and learning principles to DNN training [3, 9, 34, 56, 64, 65, 69, 72]. However, information-theoretic objectives for deterministic DNNs often face a common issue: the mutual information between the input and the DNN representation is infinite. This leads to ill-posed optimization problems.

Several strategies have been suggested to address this challenge. One involves using stochastic DNNs with variational bounds, where the output of the deterministic network is used as the parameters of the conditional distribution [43, 61]. Another approach, as suggested by Dubois et al. [22], assumes that the randomness of data augmentation among the two views is the primary source of stochasticity in the network. However, these methods assume that randomness comes from the DNN, contrary to common practice. Other research has presumed a random input but has made no assumptions about the network's representation distribution properties. Instead, it relies on general lower bounds to analyze the objective [71, 75].

## 3 Self-Supervised Learning in DNNs: An Information-Theoretic Perspective

To analyze information within deterministic networks, we first need to establish an information-theoretic perspective on SSL (Section 3.1). Subsequently, we utilize the Data Distribution Hypothesis (Section 3.2) to demonstrate its applicability to deterministic SSL networks.

### 3.1 Self-Supervised Learning from an Information-Theoretic Viewpoint

Our discussion begins with the *MultiView InfoMax principle*, which aims to maximize the mutual information $I(Z; X')$ between a view $X'$ and the second representation $Z$. As demonstrated in Federici et al. [24], we can optimize this information by employing the following lower bound:

$$I(Z, X') = H(Z) - H(Z|X') \geq H(Z) + \mathbb{E}_{x'}[\log q(z|x')]. \tag{2}$$

Here, $H(Z)$ represents the entropy of $Z$. In supervised learning, the labels $Y$ remain fixed, making the entropy term $H(Y)$ a constant. Consequently, the optimization is solely focused on the log-loss, $\mathbb{E}_{x'}[\log q(z|x')]$, which could be either cross-entropy or square loss.

However, for joint embedding networks, a degenerate solution can emerge, where all outputs "collapse" into an undesired value [16]. Upon examining Equation (2), we observe that the entropies are not fixed and can be optimized. As a result, minimizing the log loss alone can lead the representations to collapse into a trivial solution and must be regularized.

## 3.2 Understanding the Data Distribution Hypothesis

Previously, we mentioned that a naive analysis might suggest that the information in deterministic DNNs is infinite. To address this point, we investigate whether assuming a dataset is a mixture of Gaussians with non-overlapping support can provide a manageable distribution over the neural network outputs. This assumption is less restrictive compared to assuming that the neural network itself is stochastic, as it concerns the generative process of the data, not the model and training process. For a detailed discussion about the limitations of assuming stochastic networks and a comparison between stochastic networks vs stochastic input, see Appendix N. In Section 4.2, we verify that this assumption holds for real-world datasets.

The so-called manifold hypothesis allows us to treat any point as a Gaussian random variable with a low-rank covariance matrix aligned with the data's manifold tangent space [25], which enables us to examine the conditioning of a latent representation with respect to the mean of the observation, i.e., $X|\boldsymbol{x}^* \sim \mathcal{N}(\boldsymbol{x}; \boldsymbol{x}^*, \Sigma_{\boldsymbol{x}^*})$. Here, the eigenvectors of $\Sigma_{\boldsymbol{x}^*}$ align with the tangent space of the data manifold at $\boldsymbol{x}^*$, which varies with the position of $\boldsymbol{x}^*$ in space. In this setting, a dataset is considered a collection of distinct points $\boldsymbol{x}_n^*, n = 1, ..., N$, and the full data distribution is expressed as a sum of Gaussian densities with low-rank covariance, defined as:

$$X \sim \sum_{n=1}^{N} \mathcal{N}(\boldsymbol{x}_n^*, \Sigma_{\boldsymbol{x}_n^*})^{\mathbb{I}\{T=n\}} \quad \text{with} \quad T \sim \text{Cat}(N). \tag{3}$$

Here, $T$ is a uniform Categorical random variable. For simplicity, we assume that the effective support of $\mathcal{N}(\boldsymbol{x}_i^*, \Sigma_{\boldsymbol{x}_i^*})$ do not overlap (for empirical validation of this assumption see Section 4.2). The effective support is defined as $\{x \in \mathbb{R}^D : p(x) > \epsilon\}$. We can then approximate the density function as follows:

$$p(\boldsymbol{x}) \approx \mathcal{N}\left(\boldsymbol{x}; \boldsymbol{x}_{n(\boldsymbol{x})}^*, \Sigma_{\boldsymbol{x}_{n(\boldsymbol{x})}^*}\right)/N, \tag{4}$$

where $\mathcal{N}(\boldsymbol{x}; ., .)$ is the Gaussian density at $\boldsymbol{x}$ and $n(\boldsymbol{x}) = \arg\min_n (\boldsymbol{x} - \boldsymbol{x}_n^*)^T \Sigma_{\boldsymbol{x}_n^*}(\boldsymbol{x} - \boldsymbol{x}_n^*)$.

## 3.3 Data Distribution Under the Deep Neural Network Transformation

Let us consider an affine spline operator $f$, as illustrated in Section 2.1, which maps a space of dimension $D$ to a space of dimension $K$, where $K \geq D$. The image or the span of this mapping is expressed as follows:

$$\text{Im}(f) \triangleq \{f(\boldsymbol{x}) : \boldsymbol{x} \in \mathbb{R}^D\} = \bigcup_{\omega \in \Omega} \text{Aff}(\omega; \boldsymbol{A}_\omega, \boldsymbol{b}_\omega) \tag{5}$$

In this equation, $\text{Aff}(\omega; \boldsymbol{A}_\omega, \boldsymbol{b}_\omega) = \{\boldsymbol{A}_\omega \boldsymbol{x} + \boldsymbol{b}_\omega : \boldsymbol{x} \in \omega\}$ denotes the affine transformation of region $\omega$ by the per-region parameters $\boldsymbol{A}_\omega, \boldsymbol{b}_\omega$. $\Omega$ denotes the partition of the input space where $\boldsymbol{x}$ resides. To practically compute the per-region affine mapping, we set $\boldsymbol{A}_\omega$ to the Jacobian matrix of the network at the corresponding input $x$, and $b$ to be defined as $f(x) - \boldsymbol{A}_\omega x$. Therefore, the DNN mapping composed of affine transformations on each input space partition region $\omega \in \Omega$ based on the coordinate change induced by $\boldsymbol{A}_\omega$ and the shift induced by $\boldsymbol{b}_\omega$.

When the input space is associated with a density distribution, this density is transformed by the mapping $f$. Calculating the density of $f(X)$ is generally intractable. However, under the disjoint support assumption from Section 3.2, we can arbitrarily increase the density's representation power by raising the number of prototypes $N$. As a result, each Gaussian's support is contained within the region $\omega$ where its means lie, leading to the following theorem:

*Theorem* 1. Given the setting of Equation (4), the unconditional DNN output density, $Z$, can be approximated as a mixture of the affinely transformed distributions $\boldsymbol{x}|\boldsymbol{x}_{n(\boldsymbol{x})}^*$:

$$Z \sim \sum_{n=1}^{N} \mathcal{N}\left(\boldsymbol{A}_{\omega(\boldsymbol{x}_n^*)}\boldsymbol{x}_n^* + \boldsymbol{b}_{\omega(\boldsymbol{x}_n^*)}, \boldsymbol{A}_{\omega(\boldsymbol{x}_n^*)}^T \Sigma_{\boldsymbol{x}_n^*} \boldsymbol{A}_{\omega(\boldsymbol{x}_n^*)}\right)^{\mathbb{1}\{T=n\}},$$

where $\omega(\boldsymbol{x}_n^*) = \omega \in \Omega \iff \boldsymbol{x}_n^* \in \omega$ is the partition region in which the prototype $\boldsymbol{x}_n^*$ lives in.

*Proof.* See Appendix B. □

In other words, Theorem 1 implies that when the input noise is small, we can simplify the conditional output density to a single Gaussian: $(Z'|X' = x_n) \sim \mathcal{N}(\mu(x_n), \Sigma(x_n))$, where $\mu(x_n) = \boldsymbol{A}_{\omega(\boldsymbol{x}_n)}\boldsymbol{x}_n + \boldsymbol{b}_{\omega(\boldsymbol{x}_n)}$ and $\Sigma(x_n) = \boldsymbol{A}_{\omega(\boldsymbol{x}_n)}^T \Sigma_{\boldsymbol{x}_n} \boldsymbol{A}_{\omega(\boldsymbol{x}_n)}$.

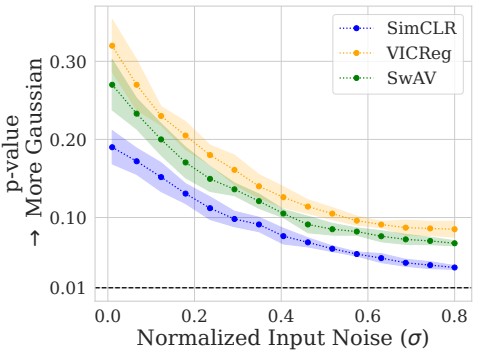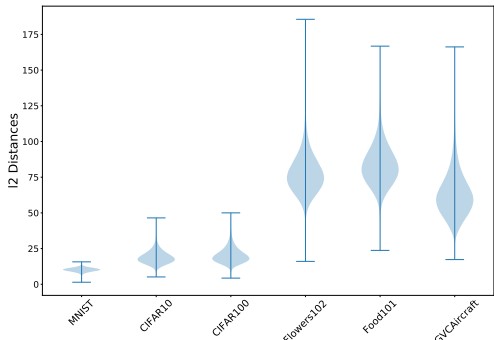

Figure 1: **Left: The network output for SSL training is more Gaussian for small input noise**. The $p$-value of the normality test for different SSL models trained on ImageNet for different input noise levels. The dashed line represents the point at which the null hypothesis (Gaussian distribution) can be rejected with $99\%$ confidence. **Right: The Gaussians around each point are not overlapping.** The plots show the $l2$ distances between raw images for different datasets. As can be seen, the distances are largest for more complex real-world datasets.

## 4 Information Optimization and the VICReg Optimization Objective

Building on our earlier discussion, we used the Data Distribution Hypothesis to model the conditional output in deterministic networks as a Gaussian mixture. This allowed us to frame the SSL training objective as maximizing the mutual information, $I(Z; X')$ and $I(Z'; X)$.

However, in general, this mutual information is intractable. Therefore, we will use our derivation for the network's representation to obtain a tractable variational approximation using the expected loss, which we can optimize.

The computation of expected loss requires us to marginalize the stochasticity in the output. We can employ maximum likelihood estimation with a Gaussian observation model. For computing the expected loss over $x'$ samples, we must marginalize the stochasticity in $Z'$. This procedure implies that the conditional decoder adheres to a Gaussian distribution: $(Z|X' = x_n) \sim \mathcal{N}(\mu(x_n), I + \Sigma(x_n))$.

However, calculating the expected log loss over samples of $Z$ is challenging. We thus focus on a lower bound – the expected log loss over $Z'$ samples. Utilizing Jensen's inequality, we derive the following lower bound:

$$\mathbb{E}_{x'}\left[\log q(z|x')\right] \geq \mathbb{E}_{z'|x'}\left[\log q(z|z')\right] = \frac{1}{2}(d\log 2\pi - (z - \mu(x'))^2 - \mathrm{Tr}\log \Sigma(x')). \quad (6)$$

Taking the expectation over $Z$, we get

$$\mathbb{E}_{z|x}\left[\mathbb{E}_{z'|x'}\left[\log q(z|z')\right]\right] = \frac{1}{2}(d\log 2\pi - (\mu(x) - \mu(x'))^2 - \log(|\Sigma(x)| \cdot |\Sigma(x')|)). \quad (7)$$

Combining all of the above, we obtain

$$I(Z; X') \geq H(Z) + \frac{d}{2}\log 2\pi - \frac{1}{2}\mathbb{E}_{x,x'}[(\mu(x) - \mu(x'))^2 + \log(|\Sigma(x)| \cdot |\Sigma(x')|)]. \quad (8)$$

The full derivations are presented in Appendix A. To optimize this objective in practice, we approximate $p(x, x')$ using the empirical data distribution

$$L(x_1 \ldots x_N, x'_1 \ldots x'_N) \approx \frac{1}{N}\sum_{i=1}^{N}\underbrace{H(Z) - \log(|\Sigma(x_i)| \cdot |\Sigma(x'_i)|)}_{\text{Regularizer}} - \underbrace{\frac{1}{2}(\mu(x_i) - \mu(x'_i))^2}_{\text{Invariance}}. \quad (9)$$

### 4.1 Variance-Invariance-Covariance Regularization: An Information-Theoretic Perspective

Next, we connect the VICReg to our information-theoretic-based objective. The "invariance term" in Equation (9), which pushes augmentations from the same image closer together, is the same term used in the VICReg objective. However, the computation of the regularization term poses a significant challenge. Entropy estimation is a well-established problem within information theory, with Gaussian mixture densities often used for representation. Yet, the differential entropy of Gaussian mixtures lacks a closed-form solution [55].

A straightforward method for approximating entropy involves capturing the distribution's first two moments, which provides an upper bound on the entropy. However, minimizing an upper bound doesn't necessarily optimize the original objective. Despite reported success from minimizing an upper bound [46, 53], this approach may induce instability during the training process.

Let $\Sigma_Z$ denote the covariance matrix of $Z$. We utilize the first two moments to approximate the entropy we aim to maximize. Because the invariance term appears in the same form as the original VICReg objective, we will look only at the regularizer. Consequently, we get the approximation

$$\bar{L}(x_1 \ldots x_N, x'_1 \ldots x'_N) \approx \sum_{i=1}^{N} \log \frac{|\Sigma_Z(x_1 \ldots x_N)|}{|\Sigma(x_i)| \cdot |\Sigma(x'_i)|}. \tag{10}$$

*Theorem* 2. Assuming that the eigenvalues of $\Sigma(x_i)$ and $\Sigma(x'_i)$, along with the differences between the Gaussian means $\mu(x_i)$ and $\mu(x'_i)$, are bounded, the solution to the maximization problem

$$\max_{\Sigma_Z} \left\{ \sum_{i=1}^{N} \log \frac{|\Sigma_Z(x_1 \ldots x_N)|}{|\Sigma(x_i)| \cdot |\Sigma(x'_i)|} \right\} \tag{11}$$

involves setting $\Sigma_Z$ to a diagonal matrix.

*Proof.* See Appendix J. $\qquad\qquad\qquad\qquad\qquad\qquad\qquad\qquad\qquad\qquad\qquad\qquad\qquad\square$

According to Theorem 2, we can maximize Equation (10) by diagonalizing the covariance matrix and increasing its diagonal elements. This goal can be achieved by minimizing the off-diagonal elements of $\Sigma_Z$–the covariance criterion of VICReg–and by maximizing the sum of the log of its diagonal elements. While this approach is straightforward and efficient, it does have a drawback: the diagonal values could tend towards zero, potentially causing instability during logarithm computations. A solution to this issue is to use an upper bound and directly compute the sum of the diagonal elements, resulting in the variance term of VICReg. This establishes the link between our information-theoretic objective and the three key components of VICReg.

### 4.2 Empirical Validation of Assumptions About Data Distributions

Validating our theory, we tested if the conditional output density $P(Z|X)$ becomes a Gaussian as input noise lessens. We used a ResNet-50 model trained with SimCLR or VICReg objectives on CIFAR-10, CIFAR-100, and ImageNet datasets. We sampled 512 Gaussian samples for each image from the test dataset, examining whether each sample remained Gaussian in the DNN's penultimate layer. We applied the D'Agostino and Pearson's test to ascertain the validity of this assumption [19].

Figure 1 (left) displays the $p$-value as a function of the normalized std. For low noise levels, we reject that the network's conditional output density is non-Gaussian with an 85% probability when using VICReg. However, the network output deviates from Gaussian as the input noise increases.

Next, we verified our assumption of non-overlapping effective support in the model's data distribution. We calculate the distribution of pairwise $l_2$ distances between images across seven datasets: MNIST [41], CIFAR10, CIFAR100 [40], Flowers102 [52], Food101 [12], and FGVAircaft [45]. Figure 1 (right) reveals that the pairwise distances are far from zero, even for raw pixels. This implies that we can use a small Gaussian around each point without overlap, validating our assumption as realistic.

# 5 Self-Supervised Learning Models through Information Maximization

The practical application of Equation (8) involves several key "design choices". We begin by comparing how existing SSL models have implemented it, investigating the estimators used, and discussing the implications of their assumptions. Subsequently, we introduce new methods for SSL that incorporate sophisticated estimators from the field of information theory, which outperform current approaches.

## 5.1 VICReg vs. SimCLR

In order to evaluate their underlying assumptions and strategies for information maximization, we compare VICReg to contrastive SSL methods such as SimCLR along with non-contrastive methods like BYOL and SimSiam.

**Contrastive Learning with SimCLR.** In their work, Lee et al. [43] drew a connection between the SimCLR objective and the variational bound on information regarding representations by employing the von Mises-Fisher distribution. By applying our analysis for information in deterministic networks with their work, we compare the main differences between SimCLR and VICReg:

(i) **Conditional distribution:** SimCLR assumes a von Mises-Fisher distribution for the encoder, while VICReg assumes a Gaussian distribution. (ii) **Entropy estimation:** SimCLR approximated it based on the finite sum of the input samples. In contrast, VICReg estimates the entropy of $Z$ solely based on the second moment. Developing SSL methods that integrate these two distinctions form an intriguing direction for future research.

**Empirical comparison.** We trained ResNet-18 on CIFAR-10 for VICReg, SimCLR, and BYOL and compared their entropies directly using the *pairwise distances* entropy estimator. (For more details, see Appendix K.) This estimator was not directly optimized by any method and was an independent validation. The results (Figure 2), showed that entropy increased for all methods during training, with SimCLR having the lowest and VICReg the highest entropy.

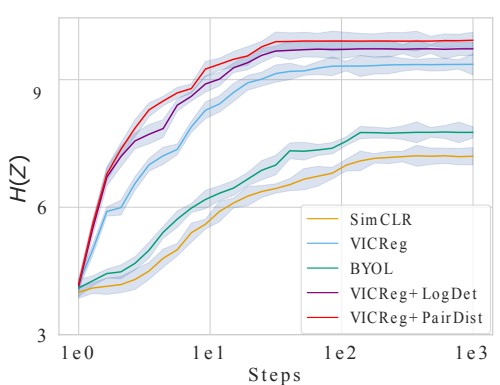

Figure 2: **VICReg has higher Entropy during training.** The entropy along the training for different SSL methods. Experiments were conducted with ResNet-18 on CIFAR-10. Error bars represent one standard error over 5 trials.

## 5.2 Family of alternative Entropy Estimators

Next, we suggest integrating the invariance term of current SSL methods with plug-in methods that optimize entropy.

**Entropy estimators.** The VICReg objective seeks to approximate the log determinant of the empirical covariance matrix through its diagonal terms. As discussed in Section 4.1, this approach has its drawbacks. An alternative is to employ different entropy estimators. The LogDet Entropy Estimator [74] is one such option, offering a tighter upper bound. This estimator employs the differential entropy of $\alpha$ order with scaled noise and has been previously shown to be a tight estimator for high-dimensional features, proving robust to random noise. However, since this estimator provides an upper bound on entropy, maximizing this bound doesn't guarantee optimization of the original objective. To counteract this, we also introduce a lower bound estimator based on the *pairwise distances* of individual mixture components [38]. In this family, a function determining pairwise distances between component densities is designated for each member. These estimators are computationally efficient and typically straightforward to optimize. For additional entropy estimators, see Appendix F. Beyond VICReg, these methods can serve as plug-in estimators for numerous SSL algorithms. Apart from VICReg, we also conducted experiments integrating these estimators with the BYOL algorithm.

Table 1: **The proposed entropy estimators outperform previous methods.** CIFAR-10, CIFAR-100, and Tiny-ImageNet top-1 accuracy under linear evaluation using ResNet-18, ConvNetX and VIT as backbones. Error bars correspond to one standard error over three trials.

| Method | CIFAR-10 ResNet-18 | Tiny-ImageNet | | CIFAR-100 | |
|---|---|---|---|---|---|
| | | ConvNetX | VIT | ConvNetX | VIT |
| SimCLR | $89.72 \pm 0.05$ | $50.86 \pm 0.13$ | $51.16 \pm 0.13$ | $67.21 \pm 0.24$ | $67.31 \pm 0.18$ |
| Barlow Twins | $88.81 \pm 0.10$ | $51.34 \pm 0.10$ | $51.40 \pm 0.16$ | $68.54 \pm 0.15$ | $68.02 \pm 0.12$ |
| SwAV | $89.12 \pm 0.13$ | $50.76 \pm 0.14$ | $51.54 \pm 0.20$ | $68.93 \pm 0.14$ | $67.89 \pm 0.21$ |
| MoCo | $89.46 \pm 0.08$ | $52.36 \pm 0.21$ | $53.06 \pm 0.21$ | $70.32 \pm 0.15$ | $69.89 \pm 0.14$ |
| VICReg | $89.32 \pm 0.09$ | $51.02 \pm 0.26$ | $52.12 \pm 0.25$ | $70.09 \pm 0.20$ | $70.12 \pm 0.17$ |
| BYOL | $89.21 \pm 0.11$ | $52.24 \pm 0.17$ | $53.44 \pm 0.20$ | $70.01 \pm 0.27$ | $69.59 \pm 0.22$ |
| VICReg + PairDist (**ours**) | $\mathbf{90.37 \pm 0.09}$ | $52.61 \pm 0.15$ | $53.70 \pm 0.13$ | $71.10 \pm 0.16$ | $70.50 \pm 0.19$ |
| VICReg + LogDet (**ours**) | $90.27 \pm 0.08$ | $52.91 \pm 0.17$ | $\mathbf{54.89 \pm 0.20}$ | $71.23 \pm 0.18$ | $70.61 \pm 0.17$ |
| BYOL + PairDist (**ours**) | $90.19 \pm 0.14$ | $\mathbf{53.47 \pm 0.22}$ | $54.33 \pm 0.21$ | $\mathbf{71.39 \pm 0.25}$ | $\mathbf{71.09 \pm 0.24}$ |
| BYOL + LogDet (**ours**) | $90.11 \pm 0.16$ | $53.19 \pm 0.25$ | $54.67 \pm 0.27$ | $71.20 \pm 0.21$ | $70.79 \pm 0.26$ |

**Setup.** Experiments were conducted on three image datasets: CIFAR-10, CIFAR-100 [39], and Tiny-ImageNet [20]. For CIFAR-10, ResNet-18 [31] was used. In contrast, both ConvNeXt [44] and Vision Transformer [21] were used for CIFAR-100 and Tiny-ImageNet. For comparison, we examined the following SSL methods: VICReg, SimCLR, BYOL, SwAV [14], Barlow Twins [73], and MoCo [33]. The quality of representation was assessed through linear evaluation. A detailed description of different methods can be found in Appendix H.

**Results.** As evidenced by Table 1, the proposed entropy estimators surpass the original SSL methods. Using a more precise entropy estimator enhances the performance of both VICReg and BYOL, compared to their initial implementations. Notably, the pairwise distance estimator, being a lower bound, achieves superior results, resonating with the theoretical preference for maximizing a true entropy's lower bound. Our findings suggest that the astute choice of entropy estimators, guided by our framework, paves the way for enhanced performance.

## 6 A Generalization Bound for Downstream Tasks

In earlier sections, we linked information theory principles with the VICReg objective. Now, we aim to extend this link to downstream generalization via a generalization bound. This connection further aligns VICReg's generalization with information maximization and implicit regularization.

**Notation.** Consider input points $x$, outputs $y \in \mathbb{R}^r$, labeled training data $S = ((x_i, y_i))_{i=1}^n$ of size $n$ and unlabeled training data $\bar{S} = ((x_i^+, x_i^{++}))_{i=1}^m$ of size $m$, where $x_i^+$ and $x_i^{++}$ share the same (unknown) label. With the unlabeled training data, we define the invariance loss

$$I_{\bar{S}}(f_\theta) = \frac{1}{m} \sum_{i=1}^m \| f_\theta(x_i^+) - f_\theta(x_i^{++}) \|,$$

where $f_\theta$ is the trained representation on the unlabeled data $\bar{S}$. We define a labeled loss $\ell_{x,y}(w) = \| W f_\theta(x) - y \|$ where $w = \text{vec}[W] \in \mathbb{R}^{dr}$ is the vectorization of the matrix $W \in \mathbb{R}^{r \times d}$. Let $w_S = \text{vec}[W_S]$ be the minimum norm solution as $W_S = \text{minimize}_{W'} \| W' \|_F$ such that

$$W' \in \arg \min_W \frac{1}{n} \sum_{i=1}^n \| W f_\theta(x_i) - y_i \|^2.$$

We also define the representation matrices

$$Z_S = [f(x_1), \dots, f(x_n)] \in \mathbb{R}^{d \times n} \quad \text{and} \quad Z_{\bar{S}} = [f(x_1^+), \dots, f(x_m^+)] \in \mathbb{R}^{d \times m},$$

and the projection matrices

$$\mathbf{P}_{Z_S} = I - Z_S^\top (Z_S Z_S^\top)^\dagger Z_S \quad \text{and} \quad \mathbf{P}_{Z_{\bar{S}}} = I - Z_{\bar{S}}^\top (Z_{\bar{S}} Z_{\bar{S}}^\top)^\dagger Z_{\bar{S}}.$$

We define the label matrix $Y_S = [y_1, \ldots, y_n]^\top \in \mathbb{R}^{n \times r}$ and the unknown label matrix $Y_{\bar{S}} = [y_1^+, \ldots, y_m^+]^\top \in \mathbb{R}^{m \times r}$, where $y_i^+$ is the unknown label of $x_i^+$. Let $\mathcal{F}$ be a hypothesis space of $f_\theta$. For a given hypothesis space $\mathcal{F}$, we define the normalized Rademacher complexity

$$\tilde{\mathcal{R}}_m(\mathcal{F}) = \frac{1}{\sqrt{m}} \mathbb{E}_{\bar{S}, \xi} \left[ \sup_{f \in \mathcal{F}} \sum_{i=1}^m \xi_i \| f(x_i^+) - f(x_i^{++}) \| \right],$$

where $\xi_1, \ldots, \xi_m$ are independent uniform random variables in $\{-1, 1\}$. It is normalized such that $\tilde{\mathcal{R}}_m(\mathcal{F}) = O(1)$ as $m \to \infty$.

## 6.1 A Generalization Bound for Variance-Invariance-Covariance Regularization

Now we will show that the VICReg objective improves generalization on supervised downstream tasks. More specifically, minimizing the unlabeled invariance loss while controlling the covariance $Z_{\bar{S}} Z_{\bar{S}}^\top$ and the complexity of representations $\tilde{\mathcal{R}}_m(\mathcal{F})$ minimizes the expected *labeled loss*:

*Theorem* 3. (Informal version). For any $\delta > 0$, with probability at least $1 - \delta$,

$$\mathbb{E}_{x,y}[\ell_{x,y}(w_S)] \le I_{\bar{S}}(f_\theta) + \frac{2}{\sqrt{m}} \| \mathbf{P}_{Z_{\bar{S}}} Y_{\bar{S}} \|_F + \frac{1}{\sqrt{n}} \| \mathbf{P}_{Z_S} Y_S \|_F + \frac{2 \tilde{\mathcal{R}}_m(\mathcal{F})}{\sqrt{m}} + \mathcal{Q}_{m,n}, \quad (12)$$

where $\mathcal{Q}_{m,n} = O(G\sqrt{\ln(1/\delta)/m} + \sqrt{\ln(1/\delta)/n}) \to 0$ as $m, n \to \infty$. In $\mathcal{Q}_{m,n}$, the value of $G$ for the term decaying at the rate $1/\sqrt{m}$ depends on the hypothesis space of $f_\theta$ and $w$ whereas the term decaying at the rate $1/\sqrt{n}$ is independent of any hypothesis space.

*Proof.* The complete version of Theorem 3 and its proof are presented in Appendix I. □

The term $\| \mathbf{P}_{Z_{\bar{S}}} Y_{\bar{S}} \|_F$ in Theorem 3 contains the unobservable label matrix $Y_{\bar{S}}$. However, we can minimize this term by using $\| \mathbf{P}_{Z_{\bar{S}}} Y_{\bar{S}} \|_F \le \| \mathbf{P}_{Z_{\bar{S}}} \|_F \| Y_{\bar{S}} \|_F$ and by minimizing $\| \mathbf{P}_{Z_{\bar{S}}} \|_F$. The factor $\| \mathbf{P}_{Z_{\bar{S}}} \|_F$ is minimized when the rank of the covariance $Z_{\bar{S}} Z_{\bar{S}}^\top$ is maximized. This can be enforced by maximizing the diagonal entries while minimizing the off-diagonal entries, as is done in VICReg.

The term $\| \mathbf{P}_{Z_S} Y_S \|_F$ contains only observable variables, and we can directly measure the value of this term using training data. In addition, the term $\| \mathbf{P}_{Z_S} Y_S \|_F$ is also minimized when the rank of the covariance $Z_S Z_S^\top$ is maximized. Since the covariances $Z_S Z_S^\top$ and $Z_{\bar{S}} Z_{\bar{S}}^\top$ concentrate to each other via concentration inequalities with the error in the order of $O(\sqrt{(\ln(1/\delta))/n} + \tilde{\mathcal{R}}_m(\mathcal{F})\sqrt{(\ln(1/\delta))/m})$, we can also minimize the upper bound on $\| \mathbf{P}_{Z_S} Y_S \|_F$ by maximizing the diagonal entries of $Z_{\bar{S}} Z_{\bar{S}}^\top$ while minimizing its off-diagonal entries, as is done in VICReg.

Thus, VICReg can be understood as a method to minimize the generalization bound in Theorem 3 by minimizing the invariance loss while controlling the covariance to minimize the *label-agnostic* upper bounds on $\| \mathbf{P}_{Z_{\bar{S}}} Y_{\bar{S}} \|_F$ and $\| \mathbf{P}_{Z_S} Y_S \|_F$. If we know *partial* information about the label $Y_{\bar{S}}$ of the unlabeled data, we can use it to minimize $\| \mathbf{P}_{Z_{\bar{S}}} Y_{\bar{S}} \|_F$ and $\| \mathbf{P}_{Z_S} Y_S \|_F$ directly.

## 6.2 Comparison of Generalization Bounds

The SimCLR generalization bound [58] requires the number of labeled classes to go infinity to close the generalization gap, whereas the VICReg bound in Theorem 3 does *not* require the number of label classes to approach infinity for the generalization gap to go to zero. This reflects that, unlike SimCLR, VICReg does not use negative pairs and thus does not use a loss function based on the implicit expectation that the labels of a negative pair are different. Another difference is that our VICReg bound improves as $n$ increases, while the previous bound of SimCLR [58] does not depend on $n$. This is because Saunshi et al. [58] assumes partial access to the true distribution per class for setting, which removes the importance of labeled data size $n$ and is not assumed in our study.

Consequently, the generalization bound in Theorem 3 provides a new insight for VICReg regarding the ratio of the effects of $m$ v.s. $n$ through $G\sqrt{\ln(1/\delta)/m} + \sqrt{\ln(1/\delta)/n}$. Finally, Theorem 3 also illuminates the advantages of VICReg over standard supervised training. That is, with standard training, the generalization bound via the Rademacher complexity requires the complexities of hypothesis spaces, $\tilde{\mathcal{R}}_n(\mathcal{W})/\sqrt{n}$ and $\tilde{\mathcal{R}}_n(\mathcal{F})/\sqrt{n}$, with respect to the size of labeled data $n$, instead of the size of unlabeled data $m$. Thus, Theorem 3 shows that using SSL, we can replace the complexities of hypothesis spaces in terms of $n$ with those in terms of $m$. Since the number of unlabeled data points is typically much larger than the number of labeled data points, this illuminates the benefit of SSL.

### 6.3 Understanding Theorem 2 via Mutual Information Maximization

Theorem 3, together with the result of the previous section, shows that, for generalization in the downstream task, it is helpful to maximize the mutual information $I(Z; X')$ in SSL via minimizing the invariance loss $I_{\bar{S}}(f_\theta)$ while controlling the covariance $Z_{\bar{S}}Z_{\bar{S}}^\top$. The term $2\tilde{\mathcal{R}}_m(\mathcal{F})/\sqrt{m}$ captures the importance of controlling the complexity of the representations $f_\theta$. To understand this term further, let us consider a discretization of the parameter space of $\mathcal{F}$ to have finite $|\mathcal{F}| < \infty$. Then, by Massart's Finite Class Lemma, we have that $\tilde{\mathcal{R}}_m(\mathcal{F}) \le C\sqrt{\ln|\mathcal{F}|}$ for some constant $C > 0$. Moreover, Shwartz-Ziv [60] shows that we can approximate $\ln|\mathcal{F}|$ by $2^{I(Z;X)}$. Thus, in Theorem 3, the term $I_{\bar{S}}(f_\theta) + \frac{2}{\sqrt{m}}\|\mathbf{P}_{Z_{\bar{S}}}Y_{\bar{S}}\|_F + \frac{1}{\sqrt{n}}\|\mathbf{P}_{Z_S}Y_S\|_F$ corresponds to $I(Z; X')$ which we want to maximize while compressing the term of $2\tilde{\mathcal{R}}_m(\mathcal{F})/\sqrt{m}$ which corresponds to $I(Z; X)$ [23, 63, 66].

Although we can explicitly add regularization on the information to control $2\tilde{\mathcal{R}}_m(\mathcal{F})/\sqrt{m}$, it is possible that $I(Z; X|X')$ and $2\tilde{\mathcal{R}}_m(\mathcal{F})/\sqrt{m}$ are implicitly regularized via implicit bias through design choises [29, 68, 30]. Thus, Theorem 3 connects the information-theoretic understanding of VICReg with the probabilistic guarantee on downstream generalization.

## 7 Limitations

In our paper, we proposed novel methods for SSL premised on information maximization. Although our methods demonstrated superior performance on some datasets, computational constraints precluded us from testing them on larger datasets. Furthermore, our study hinges on certain assumptions that, despite rigorous validation efforts, may not hold universally. While we strive for meticulous testing and validation, it's crucial to note that some assumptions might not be applicable in all scenarios or conditions. These limitations should be taken into account when interpreting our study's results.

## 8 Conclusions

We analyzed the Variance-Invariance-Covariance Regularization for self-supervised learning through an information-theoretic lens. By transferring the stochasticity required for an information-theoretic analysis to the input distribution, we showed how the VICReg objective can be derived from information-theoretic principles, used this perspective to highlight assumptions implicit in the VICReg objective, derived a VICReg generalization bound for downstream tasks, and related it to information maximization.

Building on these findings, we introduced a new VICReg-inspired SSL objective. Our probabilistic guarantee suggests that VICReg can be further improved for the settings of partial label information by aligning the covariance matrix with the partially observable label matrix, which opens up several avenues for future work, including the design of improved estimators for information-theoretic quantities and investigations into the suitability of different SSL methods for specific data characteristics.

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

# Appendix

## Table of Contents

This appendix is organized as follows:

- In Appendix A, we provide a detailed derivation of the lower bound of Equation (8) .

- In Appendix B, we provide full proof of our Theorem 1 on the network's representation distribution.

- In Appendix C, we provide additional empirical validations. Specifically, we empirically check if the optimal solution to the information maximization problem in Section 4.1 is the diagonal matrix.

- In Appendix D, we show the collapse phenomenon under Gaussian Mixture Model (GMM) using Expectation Maximization (EM) and demonstrate how it is related to SSL and how we can prevent it.

- Appendix E provides additional details on the SimCLR method.

- Appendix F provides a detailed review of entropy estimators, their implications, assumptions, and limitations.

- In Appendix G, we provide proofs for known lemmas that we are using throughout our paper.

- In Appendix H, we provide detailed information on the hyperparameters, datasets, and architectures used in our experiments in Section 5.2.

- In Appendix I, we provide full proof of our generalization bound for downstream tasks from Section 6.

- In Appendix J, we provide full proof of the theorems for Section 4.1 on the connection between information optimization and the VICReg objective.

- Appendix K provides experimental details on experiments conducted in Section 5.1 for entropy comparison between different SSL methods.

- In Appendix L, we provide detailed information on the reproducibility of our study.

- In Appendix O, we discuss the broader impact of our work. This section explores the implications, significance, and potential applications of our findings beyond the scope of the immediate study.

# Appendix A    Lower bounds on $\mathbb{E}_{x'}\left[\log q(z|x')\right]$

In this section of the supplementary material, we present the full derivation of the lower bound on $\mathbb{E}_{x'}\left[\log q(z|x')\right]$. Because $Z'|X'$ is a Gaussian, we can write it as $Z' = \mu(x') + L(x')\epsilon$ where $\epsilon \sim \mathcal{N}(0, 1)$ and $L(x')^T L(x') = \Sigma(x')$. Now, setting $\Sigma_r = I$, will give us:

$$\mathbb{E}_{x'}\left[\log q(z|x')\right]$$
$$\geq \mathbb{E}_{z'|x'}\left[\log q(z|z')\right] \tag{13}$$

$$= \mathbb{E}_{z'|x'}\left[\frac{d}{2}\log 2\pi - \frac{1}{2}\left(z - z'\right)^T (I))^{-1}\left(z - z'\right)\right] \tag{14}$$

$$= \frac{d}{2}\log 2\pi - \frac{1}{2}\mathbb{E}_{z'|x',}\left[\left(z - z'\right)^2\right] \tag{15}$$

$$= \frac{d}{2}\log 2\pi - \frac{1}{2}\mathbb{E}_\epsilon\left[\left(z - \mu(x') - L(x')\epsilon\right)^2\right] \tag{16}$$

$$= \frac{d}{2}\log 2\pi - \frac{1}{2}\mathbb{E}_\epsilon\left[\left(z - \mu(x')\right)^2 - 2\left(z - \mu(x') * L(x')\epsilon\right) + \left(\left(L(x')\epsilon\right)^T\left(L(x')\epsilon\right)\right)\right] \tag{17}$$

$$= \frac{d}{2}\log 2\pi - \frac{1}{2}\mathbb{E}_\epsilon\left[\left(z - \mu(x')\right)^2\right] + \left(z - \mu(x')L(x')\right)\mathbb{E}_\epsilon\left[\epsilon\right] - \frac{1}{2}\mathbb{E}_\epsilon\left[\epsilon^T L(x')^T L(x')\epsilon\right] \tag{18}$$

$$= \frac{d}{2}\log 2\pi - \frac{1}{2}\left(z - \mu(x')\right)^2 - \frac{1}{2}Tr\log\Sigma(x') \tag{19}$$

where $\mathbb{E}_{x'}\left[\log q(z|x')\right] = \mathbb{E}_{x'}\left[\log\mathbb{E}_{z'|x'}\left[q(z|z')\right]\right] \geq \mathbb{E}_{z'}\left[\log q(z|z')\right]$ by Jensen's inequality, $\mathbb{E}_\epsilon[\epsilon] = 0$ and $\mathbb{E}_\epsilon\left[\epsilon\left(L(x')^T L(x')\epsilon\right)\right] = Tr\log\Sigma(x')$ by the Hutchinson's estimator.

$$\mathbb{E}_{z|x}\left[\mathbb{E}_{z'|x'}\left[\log q(z|z')\right]\right] = \mathbb{E}_{z|x}\left[\frac{d}{2}\log 2\pi - \frac{1}{2}\left(z - \mu(x')\right)^2 - \frac{1}{2}Tr\log\Sigma(x')\right] \tag{20}$$

$$= \frac{d}{2}\log 2\pi - \frac{1}{2}\mathbb{E}_{z|x}\left[\left(z - \mu(x')\right)^2\right] - \frac{1}{2}Tr\log\Sigma(x') \tag{21}$$

$$= \frac{d}{2}\log 2\pi - \frac{1}{2}\mathbb{E}_\epsilon\left[\left(\mu(x) + L(x)\epsilon - \mu(x')\right)^2\right] - \frac{1}{2}Tr\log\Sigma(x') \tag{22}$$

$$= \frac{d}{2}\log 2\pi - \frac{1}{2}\mathbb{E}_\epsilon\left[\left(\mu(x) - \mu(x')\right)^2\right] + \mathbb{E}_\epsilon\left[\left(\mu(x) - \mu(x')\right)L(x)\epsilon\right]$$
$$\qquad - \frac{1}{2}\mathbb{E}_\epsilon\left[\epsilon^T L(x)^T L(x)\epsilon\right] - \frac{1}{2}Tr\log\Sigma(x') \tag{23}$$

$$= \frac{d}{2}\log 2\pi - \frac{1}{2}\left(\mu(x) - \mu(x')\right)^2 - \frac{1}{2}Tr\log\Sigma(x) - \frac{1}{2}Tr\log\Sigma(x') \tag{24}$$

$$= \frac{d}{2}\log 2\pi - \frac{1}{2}\left(\mu(x) - \mu(x')\right)^2 - \frac{1}{2}\log\left(|\Sigma(x)| \cdot |\Sigma(x')|\right) \tag{25}$$

## Appendix B  Data Distribution after Deep Network Transformation

*Theorem* 4. Given the setting of Equation (4), the unconditional DNN output density denoted as $Z$ approximates (given the truncation of the Gaussian on its effective support that is included within a single region $\omega$ of the DN's input space partition) a mixture of the affinely transformed distributions $\boldsymbol{x}|\boldsymbol{x}^*_{n(\boldsymbol{x})}$ e.g. for the Gaussian case

$$Z \sim \sum_{n=1}^{N} \mathcal{N}\Big(\boldsymbol{A}_{\omega(\boldsymbol{x}^*_n)}\boldsymbol{x}^*_n + \boldsymbol{b}_{\omega(\boldsymbol{x}^*_n)}, \boldsymbol{A}^T_{\omega(\boldsymbol{x}^*_n)}\Sigma_{\boldsymbol{x}^*_n}\boldsymbol{A}_{\omega(\boldsymbol{x}^*_n)}\Big)^{T=n},$$

where $\omega(\boldsymbol{x}^*_n) = \omega \in \Omega \iff \boldsymbol{x}^*_n \in \omega$ is the partition region in which the prototype $\boldsymbol{x}^*_n$ lives in.

*Proof.* We know that If $\int_{\omega} p(\boldsymbol{x}|\boldsymbol{x}^*_{n(\boldsymbol{x})})d\boldsymbol{x} \approx 1$ then $f$ is linear within the effective support of $p$. Therefore, any sample from $p$ will almost surely lie within a single region $\omega \in \Omega$, and therefore the entire mapping can be considered linear with respect to $p$. Thus, the output distribution is a linear transformation of the input distribution based on the per-region affine mapping. □

## Appendix C  Additional Empirical Validation

To validate empirically Theorem 2, we checke empirically if the optimal solution for

$$\sum_i \log \frac{|\Sigma_Z|}{|\Sigma_{Z|X_i}||\Sigma_{Z'|X'_i}|}$$

is a diagonal matrix. We trained VICReg on ResNet18 on CIFAR-10 and did random perturbations (with different scales) for $\Sigma_Z$. Then, for each perturbation, we calculated the average distance of this perturbed matrix from a diagonal matrix and the actual value of the term

$$\sum_i \log \frac{|\Sigma_Z||\Sigma_{Z'|X'_i}|}{|\Sigma_{Z|X_i}|}$$

. In Figure 3, we plot the difference from the optimal value of this term as a function of the distance from the diagonal matrix. As we can see, we get an optimal solution where we are close to the diagonal matrix. This observation gives us an empirical validation of Theorem 2.

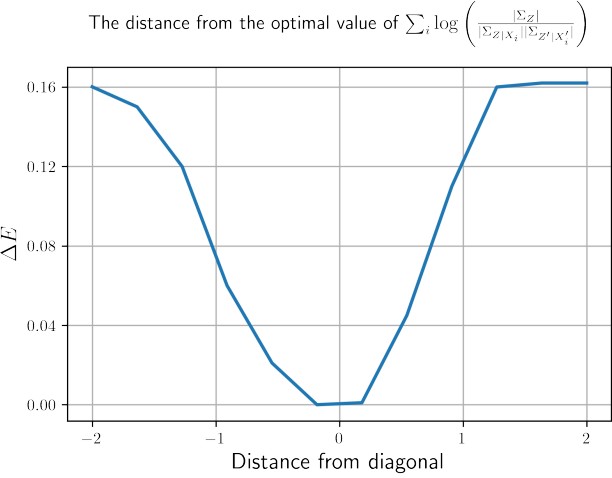

Figure 3: **The optimal solution for the optimization problem is a diagonal matrix.** The average distance from a diagonal matrix for different perturbation scales. Experiments were conducted on CIFAR-10 with the ResNet-18 network.

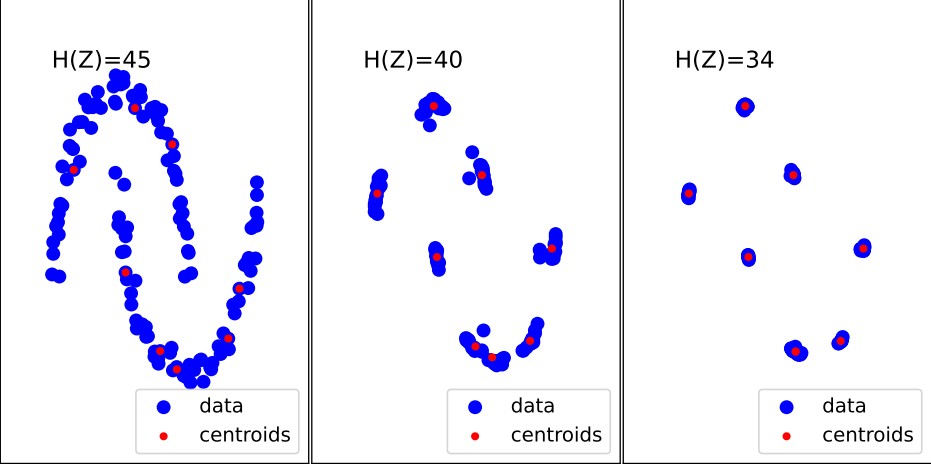

Figure 4: **Evolution of GMM training when enforcing a one-to-one mapping between the data and centroids akin to K-means i.e. using a small and fixed covariance matrix. We see that collapse does not occur.** Left - In the presence of fixed input samples, we observe that there is no collapsing and that the entropy of the centers is high. Right - when we make the input samples trainable and optimize their location, all the points collapse into a single point, resulting in a sharp decrease in entropy.

## Appendix D   EM and GMM

Let us examine a toy dataset on the pattern of two intertwining moons to illustrate the collapse phenomenon under GMM (Figure 1, right). We begin by training a classical GMM with maximum likelihood, where the means are initialized based on random samples, and the covariance is used as the identity matrix. A red dot represents the Gaussian's mean after training, while a blue dot represents the data points. In the presence of fixed input samples, we observe that there is no collapsing and that the entropy of the centers is high (Figure 4, left). However, when we make the input samples trainable and optimize their location, all the points collapse into a single point, resulting in a sharp decrease in entropy (Figure 4, right).

To prevent collapse, we follow the K-means algorithm in enforcing sparse posteriors, i.e. using small initial standard deviations and learning only the mean. This forces a one-to-one mapping which leads all points to be closest to the mean without collapsing, resulting in high entropy (Figure 4 - middle, in the Appendix). Another option to prevent collapse is to use different learning rates for input and parameters. Using this setting, the collapsing of the parameters does not maximize the likelihood. Figure 1 (right) shows the results of GMM with different learning rates for learned inputs and parameters. When the parameter learning rate is sufficiently high in comparison to the input learning rate, the entropy decreases much more slowly and no collapse occurs.

## Appendix E   SimCLR

In contrastive learning, different augmented views of the same image are attracted (positive pairs), while different augmented views are repelled (negative pairs). MoCo [33] and SimCLR [16] are recent examples of self-supervised visual representation learning that reduce the gap between self-supervised and fully-supervised learning. SimCLR applies randomized augmentations to an image to create two different views, $x$ and $y$, and encodes both of them with a shared encoder, producing representations $r_x$ and $r_y$. Both $r_x$ and $r_y$ are $l2$-normalized. The SimCLR version of the InfoNCE objective is:

$$\mathbb{E}_{x,y}\left[-\log\left(\frac{e^{\frac{1}{\eta}r_y^T r_x}}{\sum_{k=1}^{K} e^{\frac{1}{\eta}r_{y_k}^T r_x}}\right)\right],$$

where $\eta$ is a temperature term and $K$ is the number of views in a minibatch.

## Appendix F  Entropy Estimators

Entropy estimation is one of the classical problems in information theory, where Gaussian mixture density is one of the most popular representations. With a sufficient number of components, they can approximate any smooth function with arbitrary accuracy. For Gaussian mixtures, there is, however, no closed-form solution to differential entropy. There exist several approximations in the literature, including loose upper and lower bounds [35]. Monte Carlo (MC) sampling is one way to approximate Gaussian mixture entropy. With sufficient MC samples, an unbiased estimate of entropy with an arbitrarily accurate can be obtained. Unfortunately, MC sampling is very computationally expensive and typically requires a large number of samples, especially in high dimensions [13]. Using the first two moments of the empirical distribution, VIGCreg used one of the most straightforward approaches for approximating the entropy. Despite this, previous studies have found that this method is a poor approximation of the entropy in many cases [35]. Another option is to use the LogDet function. Several estimators have been proposed to implement it, including uniformly minimum variance unbiased (UMVU) [2], and Bayesian methods [49]. These methods, however, often require complex optimizations. The LogDet estimator presented in [74] used the differential entropy $\alpha$ order entropy using scaled noise. They demonstrated that it can be applied to high-dimensional features and is robust to random noise. Based on Taylor-series expansions, [35] presented a lower bound for the entropy of Gaussian mixture random vectors. They use Taylor-series expansions of the logarithm of each Gaussian mixture component to get an analytical evaluation of the entropy measure. In addition, they present a technique for splitting Gaussian densities to avoid components with high variance, which would require computationally expensive calculations. Kolchinsky and Tracey [38] introduce a novel family of estimators for the mixture entropy. For this family, a pairwise-distance function between component densities is defined for each member. These estimators are computationally efficient as long as the pairwise-distance function and the entropy of each component distribution are easy to compute. Moreover, the estimator is continuous and smooth and is therefore useful for optimization problems. In addition, they presented both a lower bound (using Chernoff distance) and an upper bound (using the KL divergence) on the entropy, which are exact when the component distributions are grouped into well-separated clusters,

## Appendix G  Known Lemmas

We use the following well-known theorems as lemmas in our proofs. We put these below for completeness. These are classical results and *not* our results.

**Lemma G.1.** (Hoeffding's inequality) *Let $X_1, ..., X_n$ be independent random variables such that $a \leq X_i \leq b$ almost surely. Consider the average of these random variables, $S_n = \dfrac{1}{n}(X_1 + \cdots + X_n)$. Then, for all $t > 0$,*

$$\mathbb{P}_S\left(\mathrm{E}\left[S_n\right] - S_n \geq (b-a)\sqrt{\frac{\ln(1/\delta)}{2n}}\right) \leq \delta,$$

*and*

$$\mathbb{P}_S\left(S_n - \mathrm{E}\left[S_n\right] \geq (b-a)\sqrt{\frac{\ln(1/\delta)}{2n}}\right) \leq \delta.$$

*Proof.* By using Hoeffding's inequality, we have that for all $t > 0$,

$$\mathbb{P}_S\left(\mathrm{E}\left[S_n\right] - S_n \geq t\right) \leq \exp\left(-\frac{2nt^2}{(b-a)^2}\right),$$

and

$$\mathbb{P}_S\left(S_n - \mathrm{E}\left[S_n\right] \geq t\right) \leq \exp\left(-\frac{2nt^2}{(b-a)^2}\right),$$

Setting $\delta = \exp\left(-\frac{2nt^2}{(b-a)^2}\right)$ and solving for $t > 0$,

$$1/\delta = \exp\left(\frac{2nt^2}{(b-a)^2}\right)$$

$$\implies \ln(1/\delta) = \frac{2nt^2}{(b-a)^2}$$

$$\implies \frac{(b-a)^2 \ln(1/\delta)}{2n} = t^2$$

$$\implies t = (b-a)\sqrt{\frac{\ln(1/\delta)}{2n}}$$

$\square$

It has been shown that generalization bounds can be obtained via Rademacher complexity [8, 50, 59]. The following is a trivial modification of [50, Theorem 3.1] for a one-sided bound on the nonnegative general loss functions:

**Lemma G.2.** *Let $\mathcal{G}$ be a set of functions with the codomain $[0, M]$. Then, for any $\delta > 0$, with probability at least $1 - \delta$ over an i.i.d. draw of $m$ samples $S = (q_i)_{i=1}^m$, the following holds for all $\psi \in \mathcal{G}$:*

$$\mathbb{E}_q[\psi(q)] \leq \frac{1}{m}\sum_{i=1}^m \psi(q_i) + 2\mathcal{R}_m(\mathcal{G}) + M\sqrt{\frac{\ln(1/\delta)}{2m}}, \tag{26}$$

*where $\mathcal{R}_m(\mathcal{G}) := \mathbb{E}_{S,\xi}[\sup_{\psi \in \mathcal{G}} \frac{1}{m}\sum_{i=1}^m \xi_i\psi(q_i)]$ and $\xi_1, \dots, \xi_m$ are independent uniform random variables taking values in $\{-1, 1\}$.*

*Proof.* Let $S = (q_i)_{i=1}^m$ and $S' = (q_i')_{i=1}^m$. Define

$$\varphi(S) = \sup_{\psi \in \mathcal{G}} \mathbb{E}_{x,y}[\psi(q)] - \frac{1}{m}\sum_{i=1}^m \psi(q_i). \tag{27}$$

To apply McDiarmid's inequality to $\varphi(S)$, we compute an upper bound on $|\varphi(S) - \varphi(S')|$ where $S$ and $S'$ be two test datasets differing by exactly one point of an arbitrary index $i_0$; i.e., $S_i = S_i'$ for all $i \neq i_0$ and $S_{i_0} \neq S_{i_0}'$. Then,

$$\varphi(S') - \varphi(S) \leq \sup_{\psi \in \mathcal{G}} \frac{\psi(q_{i_0}) - \psi(q_{i_0}')}{m} \leq \frac{M}{m}. \tag{28}$$

Similarly, $\varphi(S) - \varphi(S') \leq \frac{M}{m}$. Thus, by McDiarmid's inequality, for any $\delta > 0$, with probability at least $1 - \delta$,

$$\varphi(S) \leq \mathbb{E}_S[\varphi(S)] + M\sqrt{\frac{\ln(1/\delta)}{2m}}. \tag{29}$$

Moreover,

$$\mathbb{E}_S[\varphi(S)] = \mathbb{E}_S\left[\sup_{\psi \in \mathcal{G}} \mathbb{E}_{S'}\left[\frac{1}{m}\sum_{i=1}^m \psi(q_i')\right] - \frac{1}{m}\sum_{i=1}^m \psi(q_i)\right] \tag{30}$$

$$\leq \mathbb{E}_{S,S'}\left[\sup_{\psi \in \mathcal{G}} \frac{1}{m}\sum_{i=1}^m (\psi(q_i') - \psi(q_i))\right] \tag{31}$$

$$\leq \mathbb{E}_{\xi,S,S'}\left[\sup_{\psi \in \mathcal{G}} \frac{1}{m}\sum_{i=1}^m \xi_i(\psi(q_i') - \psi(q_i))\right] \tag{32}$$

$$\leq 2\mathbb{E}_{\xi,S}\left[\sup_{\psi \in \mathcal{G}} \frac{1}{m}\sum_{i=1}^m \xi_i\psi(q_i)\right] = 2\mathcal{R}_m(\mathcal{G}), \tag{33}$$

where the first line follows the definitions of each term, the second line uses Jensen's inequality and the convexity of the supremum, and the third line follows that for each $\xi_i \in \{-1, +1\}$, the distribution of each term $\xi_i(\ell(f(x_i'), y_i') - \ell(f(x_i), y_i))$ is the distribution of $(\ell(f(x_i'), y_i') - \ell(f(x_i), y_i))$ since $S$ and $S'$ are drawn iid with the same distribution. The fourth line uses the subadditivity of supremum. $\square$

# Appendix H   Implentation Details for Maximizing Entropy Estimators

In this section, we will provide more details on the implantation of the experiments conducted in Section 5.2.

**Setup** Our experiments are conducted on CIFAR-10 [40]. We use ResNet-18 [32] as our backbone.

**Training Procedure**: The experimental process is organized into two sequential stages: unsupervised pretraining followed by linear evaluation. Initially, the unsupervised pretraining phase is executed, during which the encoder network is trained. Upon its completion, we transition to the linear evaluation phase, which serves as an assessment tool for the quality of the representation produced by the pretrained encoder.

Once the pretraining phase is concluded, we adhere to the fine-tuning procedures used in established baseline methods, as described by [14].

During the linear evaluation stage, we start by performing supervised training of the linear classifier. This is achieved by using the representations derived from the encoder network while keeping the network's coefficients frozen, and applying the same training dataset. Subsequently, we measure the test accuracy of the trained linear classifier using a separate validation dataset. This approach allows us to evaluate the performance of our model in a robust and systematic manner.

The training process for each model unfolds over 800 epochs, employing a batch size of 512. We utilize the Stochastic Gradient Descent (SGD) optimizer, characterized by a momentum of 0.9 and a weight decay of $1e-4$. The learning rate is initiated at 0.5 and is adjusted according to a cosine decay schedule complemented by a linear warmup phase.

During the data augmentation process, two enhanced versions of every input image are generated. This involves cropping each image randomly and resizing it back to the original resolution. The images are then subject to random horizontal flipping, color jittering, grayscale conversion, Gaussian blurring, and polarization for further augmentation.

For the linear evaluation phase, the linear classifier is trained for 100 epochs with a batch size of 256. The SGD optimizer is again employed, this time with a momentum of 0.9 and no weight decay. The learning rate is managed using a cosine decay schedule, starting at 0.2 and reaching a minimum of $2e-4$.

# Appendix I   A Generalization Bound for Downstream Tasks

In this Appendix, we present the complete version of Theorem 3 along with its proof and additional discussions.

## I.1   Additional Notation and details

We start to introduce additional notation and details. We use the notation of $x \in \mathcal{X}$ for an input and $y \in \mathcal{Y} \subseteq \mathbb{R}^r$ for an output. Define $p(y) = \mathbb{P}(Y = y)$ to be the probability of getting label $y$ and $\hat{p}(y) = \frac{1}{n} \sum_{i=1}^{n} \mathbb{1}\{y_i = y\}$ to be the empirical estimate of $p(y)$. Let $\zeta$ be an upper bound on the norm of the label as $\|y\|_2 \leq \zeta$ for all $y \in \mathcal{Y}$. Define the minimum norm solution $W_{\bar{S}}$ of the unlabeled data as $W_{\bar{S}} = \text{minimize}_{W'} \|W'\|_F$ s.t. $W' \in \arg\min_W \frac{1}{m} \sum_{i=1}^{m} \|W f_\theta(x_i^+) - g^*(x_i^+)\|^2$. Let $\kappa_S$ be a data-dependent upper bound on the per-sample Euclidian norm loss with the trained model as $\|W_S f_\theta(x) - y\| \leq \kappa_S$ for all $(x, y) \in \mathcal{X} \times \mathcal{Y}$. Similarly, let $\kappa_{\bar{S}}$ be a data-dependent upper bound on the per-sample Euclidian norm loss as $\|W_{\bar{S}} f_\theta(x) - y\| \leq \kappa_{\bar{S}}$ for all $(x, y) \in \mathcal{X} \times \mathcal{Y}$. Define the difference between $W_S$ and $W_{\bar{S}}$ by $c = \|W_S - W_{\bar{S}}\|_2$. Let $\mathcal{W}$ be a hypothesis space of $W$ such that $W_{\bar{S}} \in \mathcal{W}$. We denote by $\tilde{\mathcal{R}}_m(\mathcal{W} \circ \mathcal{F}) = \frac{1}{\sqrt{m}} \mathbb{E}_{\bar{S},\xi}[\sup_{W \in \mathcal{W}, f \in \mathcal{F}} \sum_{i=1}^{m} \xi_i \|g^*(x_i^+) - W f(x_i^+)\|]$ the normalized Rademacher complexity of the set $\{x^+ \mapsto \|g^*(x^+) - W f(x^+)\| : W \in \mathcal{W}, f \in \mathcal{F}\}$. we denote by $\kappa$ a upper bound on the per-sample Euclidian norm loss as $\|W f(x) - y\| \leq \kappa$ for all $(x, y, W, f) \in \mathcal{X} \times \mathcal{Y} \times \mathcal{W} \times \mathcal{F}$.

We adopt the following data-generating process model that was used in a previous paper on analyzing contrastive learning [58, 10]. For the labeled data, first, $y$ is drawn from the distribution $\rho$ on $\mathcal{Y}$, and then $x$ is drawn from the conditional distribution $\mathcal{D}_y$ conditioned on the label $y$. That is, we have the join distribution $\mathcal{D}(x, y) = \mathcal{D}_y(x)\rho(y)$ with $((x_i, y_i))_{i=1}^{n} \sim \mathcal{D}^n$. For the unlabeled data,

first, each of the *unknown* labels $y^+$ and $y^-$ is drawn from the distribuion $\rho$, and then each of the positive examples $x^+$ and $x^{++}$ is drawn from the conditional distribution $\mathcal{D}_{y^+}$ while the negative example $x^-$ is drawn from the $\mathcal{D}_{y^-}$. Unlike the analysis of contrastive learning, we do not require negative samples. Let $\tau_{\bar{S}}$ be a data-dependent upper bound on the invariance loss with the trained representation as $\|f_\theta(\bar{x}) - f_\theta(x)\| \leq \tau_{\bar{S}}$ for all $(\bar{x}, x) \sim \mathcal{D}_y^2$ and $y \in \mathcal{Y}$. Let $\tau$ be a data-independent upper bound on the invariance loss with the trained representation as $\|f(\bar{x}) - f(x)\| \leq \tau$ for all $(\bar{x}, x) \sim \mathcal{D}_{y'}^2$, $y \in \mathcal{Y}$, and $f \in \mathcal{F}$. For simplicity, we assume that there exists a function $g^*$ such that $y = g^*(x) \in \mathbb{R}^r$ for all $(x, y) \in \mathcal{X} \times \mathcal{Y}$. Discarding this assumption adds the average of label noises to the final result, which goes to zero as the sample sizes $n$ and $m$ increase, assuming that the mean of the label noise is zero.

## 0.2  Proof of Theorem 3

*Proof of Theorem 3.* Let $W = W_S$ where $W_S$ is the the minimum norm solution as $W_S = \text{minimize}_{W'} \|W'\|_F$ s.t. $W' \in \arg\min_W \frac{1}{n} \sum_{i=1}^n \|W f_\theta(x_i) - y_i\|^2$. Let $W^* = W_{\bar{S}}$ where $W_{\bar{S}}$ is the minimum norm solution as $W^* = W_{\bar{S}} = \text{minimize}_{W'} \|W'\|_F$ s.t. $W' \in \arg\min_W \frac{1}{m} \sum_{i=1}^m \|W f_\theta(x_i^+) - g^*(x_i^+)\|^2$. Since $y = g^*(x)$,

$$y = g^*(x) \pm W^* f_\theta(x) = W^* f_\theta(x) + (g^*(x) - W^* f_\theta(x)) = W^* f_\theta(x) + \varphi(x)$$

where $\varphi(x) = g^*(x) - W^* f_\theta(x)$. Define $L_S(w) = \frac{1}{n} \sum_{i=1}^n \|W f_\theta(x_i) - y_i\|$. Using these,

$$L_S(w) = \frac{1}{n} \sum_{i=1}^n \|W f_\theta(x_i) - y_i\|$$

$$= \frac{1}{n} \sum_{i=1}^n \|W f_\theta(x_i) - W^* f_\theta(x_i) - \varphi(x_i)\|$$

$$\geq \frac{1}{n} \sum_{i=1}^n \|W f_\theta(x_i) - W^* f_\theta(x_i)\| - \frac{1}{n} \sum_{i=1}^n \|\varphi(x_i)\|$$

$$= \frac{1}{n} \sum_{i=1}^n \|\tilde{W} f_\theta(x_i)\| - \frac{1}{n} \sum_{i=1}^n \|\varphi(x_i)\|$$

where $\tilde{W} = W - W^*$. We now consider new fresh samples $\bar{x}_i \sim \mathcal{D}_{y_i}$ for $i = 1, \ldots, n$ to rewrite the above further as:

$$L_S(w) \geq \frac{1}{n} \sum_{i=1}^n \|\tilde{W} f_\theta(x_i) \pm \tilde{W} f_\theta(\bar{x}_i)\| - \frac{1}{n} \sum_{i=1}^n \|\varphi(x_i)\|$$

$$= \frac{1}{n} \sum_{i=1}^n \|\tilde{W} f_\theta(\bar{x}_i) - (\tilde{W} f_\theta(\bar{x}_i) - \tilde{W} f_\theta(x_i))\| - \frac{1}{n} \sum_{i=1}^n \|\varphi(x_i)\|$$

$$\geq \frac{1}{n} \sum_{i=1}^n \|\tilde{W} f_\theta(\bar{x}_i)\| - \frac{1}{n} \sum_{i=1}^n \|\tilde{W} f_\theta(\bar{x}_i) - \tilde{W} f_\theta(x_i)\| - \frac{1}{n} \sum_{i=1}^n \|\varphi(x_i)\|$$

$$= \frac{1}{n} \sum_{i=1}^n \|\tilde{W} f_\theta(\bar{x}_i)\| - \frac{1}{n} \sum_{i=1}^n \|\tilde{W}(f_\theta(\bar{x}_i) - f_\theta(x_i))\| - \frac{1}{n} \sum_{i=1}^n \|\varphi(x_i)\|$$

This implies that

$$\frac{1}{n} \sum_{i=1}^n \|\tilde{W} f_\theta(\bar{x}_i)\| \leq L_S(w) + \frac{1}{n} \sum_{i=1}^n \|\tilde{W}(f_\theta(\bar{x}_i) - f_\theta(x_i))\| + \frac{1}{n} \sum_{i=1}^n \|\varphi(x_i)\|.$$

Furthermore, since $y = W^* f_\theta(x) + \varphi(x)$, by writing $\bar{y}_i = W^* f_\theta(\bar{x}_i) + \varphi(\bar{x}_i)$ (where $\bar{y}_i = y_i$ since $\bar{x}_i \sim \mathcal{D}_{y_i}$ for $i = 1, \ldots, n$),

$$\frac{1}{n} \sum_{i=1}^n \|\tilde{W} f_\theta(\bar{x}_i)\| = \frac{1}{n} \sum_{i=1}^n \|W f_\theta(\bar{x}_i) - W^* f_\theta(\bar{x}_i)\|$$

$$= \frac{1}{n} \sum_{i=1}^n \|W f_\theta(\bar{x}_i) - \bar{y}_i + \varphi(\bar{x}_i)\|$$

$$\geq \frac{1}{n} \sum_{i=1}^n \|W f_\theta(\bar{x}_i) - \bar{y}_i\| - \frac{1}{n} \sum_{i=1}^n \|\varphi(\bar{x}_i)\|$$

Combining these, we have that

$$\frac{1}{n} \sum_{i=1}^n \|W f_\theta(\bar{x}_i) - \bar{y}_i\| \leq L_S(w) + \frac{1}{n} \sum_{i=1}^n \|\tilde{W} (f_\theta(\bar{x}_i) - f_\theta(x_i))\| \tag{34}$$

$$+ \frac{1}{n} \sum_{i=1}^n \|\varphi(x_i)\| + \frac{1}{n} \sum_{i=1}^n \|\varphi(\bar{x}_i)\|.$$

To bound the left-hand side of equation 34, we now analyze the following random variable:

$$\mathbb{E}_{X,Y}[\|W_S f_\theta(X) - Y\|] - \frac{1}{n} \sum_{i=1}^n \|W_S f_\theta(\bar{x}_i) - \bar{y}_i\|, \tag{35}$$

where $\bar{y}_i = y_i$ since $\bar{x}_i \sim \mathcal{D}_{y_i}$ for $i = 1, \ldots, n$. Importantly, this means that as $W_S$ depends on $y_i$, $W_S$ depends on $\bar{y}_i$. Thus, the collection of random variables $\|W_S f_\theta(\bar{x}_1) - \bar{y}_1\|, \ldots, \|W_S f_\theta(n_n) - \bar{y}_n\|$ is *not* independent. Accordingly, we cannot apply standard concentration inequality to bound equation 35. A standard approach in learning theory is to first bound equation 35 by $\mathbb{E}_{x,y}\|W_S f_\theta(x) - y\| - \frac{1}{n} \sum_{i=1}^n \|W_S f_\theta(\bar{x}_i) - \bar{y}_i\| \leq \sup_{W \in \mathcal{W}} \mathbb{E}_{x,y}\|W f_\theta(x) - y\| - \frac{1}{n} \sum_{i=1}^n \|W f_\theta(\bar{x}_i) - \bar{y}_i\|$ for some hypothesis space $\mathcal{W}$ (that is independent of $S$) and realize that the right-hand side now contains the collection of independent random variables $\|W f_\theta(\bar{x}_1) - \bar{y}_1\|, \ldots, \|W f_\theta(n_n) - \bar{y}_n\|$, for which we can utilize standard concentration inequalities. This reasoning leads to the Rademacher complexity of the hypothesis space $\mathcal{W}$. However, the complexity of the hypothesis space $\mathcal{W}$ can be very large, resulting in a loose bound. In this proof, we show that we can avoid the dependency on hypothesis space $\mathcal{W}$ by using a very different approach with conditional expectations to take care the dependent random variables $\|W_S f_\theta(\bar{x}_1) - \bar{y}_1\|, \ldots, \|W_S f_\theta(n_n) - \bar{y}_n\|$. Intuitively, we utilize the fact that for these dependent random variables, there is a structure of conditional independence, conditioned on each $y \in \mathcal{Y}$.

We first write the expected loss as the sum of the conditional expected loss:

$$\mathbb{E}_{X,Y}[\|W_S f_\theta(X) - Y\|] = \sum_{y \in \mathcal{Y}} \mathbb{E}_{X,Y}[\|W_S f_\theta(X) - Y\| \mid Y = y]\mathbb{P}(Y = y)$$

$$= \sum_{y \in \mathcal{Y}} \mathbb{E}_{X_y}[\|W_S f_\theta(X_y) - y\|]\mathbb{P}(Y = y),$$

where $X_y$ is the random variable for the conditional with $Y = y$. Using this, we decompose equation 35 into two terms:

$$\mathbb{E}_{X,Y}[\|W_S f_\theta(X) - Y\|] - \frac{1}{n} \sum_{i=1}^n \|W_S f_\theta(\bar{x}_i) - \bar{y}_i\| \tag{36}$$

$$= \left( \sum_{y \in \mathcal{Y}} \mathbb{E}_{X_y}[\|W_S f_\theta(X_y) - y\|]\frac{|\mathcal{I}_y|}{n} - \frac{1}{n} \sum_{i=1}^n \|W_S f_\theta(\bar{x}_i) - \bar{y}_i\| \right)$$

$$+ \sum_{y \in \mathcal{Y}} \mathbb{E}_{X_y}[\|W_S f_\theta(X_y) - y\|] \left( \mathbb{P}(Y = y) - \frac{|\mathcal{I}_y|}{n} \right),$$

where
$$\mathcal{I}_y = \{i \in [n] : y_i = y\}.$$
The first term in the right-hand side of equation 36 is further simplified by using
$$\frac{1}{n}\sum_{i=1}^{n}\|W_S f_\theta(\bar{x}_i) - \bar{y}_i\| = \frac{1}{n}\sum_{y\in\mathcal{Y}}\sum_{i\in\mathcal{I}_y}\|W_S f_\theta(\bar{x}_i) - y\|,$$

as

$$\sum_{y\in\mathcal{Y}}\mathbb{E}_{X_y}[\|W_S f_\theta(X_y) - y\|]\frac{|\mathcal{I}_y|}{n} - \frac{1}{n}\sum_{i=1}^{n}\|W_S f_\theta(\bar{x}_i) - \bar{y}_i\|$$

$$= \frac{1}{n}\sum_{y\in\tilde{\mathcal{Y}}}|\mathcal{I}_y|\left(\mathbb{E}_{X_y}[\|W_S f_\theta(X_y) - y\|] - \frac{1}{|\mathcal{I}_y|}\sum_{i\in\mathcal{I}_y}\|W_S f_\theta(\bar{x}_i) - y\|\right),$$

where $\tilde{\mathcal{Y}} = \{y \in \mathcal{Y} : |\mathcal{I}_y| \neq 0\}$. Substituting these into equation equation 36 yields

$$\mathbb{E}_{X,Y}[\|W_S f_\theta(X) - Y\|] - \frac{1}{n}\sum_{i=1}^{n}\|W_S f_\theta(\bar{x}_i) - \bar{y}_i\| \tag{37}$$

$$= \frac{1}{n}\sum_{y\in\tilde{\mathcal{Y}}}|\mathcal{I}_y|\left(\mathbb{E}_{X_y}[\|W_S f_\theta(X_y) - y\|] - \frac{1}{|\mathcal{I}_y|}\sum_{i\in\mathcal{I}_y}\|W_S f_\theta(\bar{x}_i) - y\|\right)$$

$$+ \sum_{y\in\mathcal{Y}}\mathbb{E}_{X_y}[\|W_S f_\theta(X_y) - y\|]\left(\mathbb{P}(Y = y) - \frac{|\mathcal{I}_y|}{n}\right)$$

Importantly, while $\|W_S f_\theta(\bar{x}_1) - \bar{y}_1\|, \dots, \|W_S f_\theta(\bar{x}_n) - \bar{y}_n\|$ on the right-hand side of equation 37 are dependent random variables, $\|W_S f_\theta(\bar{x}_1) - y\|, \dots, \|W_S f_\theta(\bar{x}_n) - y\|$ are independent random variables since $W_S$ and $\bar{x}_i$ are independent and $y$ is fixed here. Thus, by using Hoeffding's inequality (Lemma G.1), and taking union bounds over $y \in \tilde{\mathcal{Y}}$, we have that with probability at least $1 - \delta$, the following holds for all $y \in \tilde{\mathcal{Y}}$:

$$\mathbb{E}_{X_y}[\|W_S f_\theta(X_y) - y\|] - \frac{1}{|\mathcal{I}_y|}\sum_{i\in\mathcal{I}_y}\|W_S f_\theta(\bar{x}_i) - y\| \le \kappa_S\sqrt{\frac{\ln(|\tilde{\mathcal{Y}}|/\delta)}{2|\mathcal{I}_y|}}.$$

This implies that with probability at least $1 - \delta$,

$$\frac{1}{n}\sum_{y\in\tilde{\mathcal{Y}}}|\mathcal{I}_y|\left(\mathbb{E}_{X_y}[\|W_S f_\theta(X_y) - y\|] - \frac{1}{|\mathcal{I}_y|}\sum_{i\in\mathcal{I}_y}\|W_S f_\theta(\bar{x}_i) - y\|\right)$$

$$\le \frac{\kappa_S}{n}\sum_{y\in\tilde{\mathcal{Y}}}|\mathcal{I}_y|\sqrt{\frac{\ln(|\tilde{\mathcal{Y}}|/\delta)}{2|\mathcal{I}_y|}}$$

$$= \kappa_S\left(\sum_{y\in\tilde{\mathcal{Y}}}\sqrt{\frac{|\mathcal{I}_y|}{n}}\right)\sqrt{\frac{\ln(|\tilde{\mathcal{Y}}|/\delta)}{2n}}.$$

Substituting this bound into equation 37, we have that with probability at least $1 - \delta$,

$$\mathbb{E}_{X,Y}[\|W_S f_\theta(X) - Y\|] - \frac{1}{n}\sum_{i=1}^{n}\|W_S f_\theta(\bar{x}_i) - \bar{y}_i\| \tag{38}$$

$$\le \kappa_S\left(\sum_{y\in\tilde{\mathcal{Y}}}\sqrt{\hat{p}(y)}\right)\sqrt{\frac{\ln(|\tilde{\mathcal{Y}}|/\delta)}{2n}} + \sum_{y\in\mathcal{Y}}\mathbb{E}_{X_y}[\|W_S f_\theta(X_y) - y\|]\left(\mathbb{P}(Y = y) - \frac{|\mathcal{I}_y|}{n}\right)$$

where

$$\hat{p}(y) = \frac{|\mathcal{I}_y|}{n}.$$

Moreover, for the second term on the right-hand side of equation 38, by using Lemma 1 of [37], we have that with probability at least $1 - \delta$,

$$\sum_{y \in \mathcal{Y}} \mathbb{E}_{X_y}[\|W_S f_\theta(X_y) - y\|] \left( \mathbb{P}(Y = y) - \frac{|\mathcal{I}_y|}{n} \right)$$

$$\leq \left( \sum_{y \in \mathcal{Y}} \sqrt{p(y)} \mathbb{E}_{X_y}[\|W_S f_\theta(X_y) - y\|] \right) \sqrt{\frac{2 \ln(|\mathcal{Y}|/\delta)}{2n}}$$

$$\leq \kappa_S \left( \sum_{y \in \mathcal{Y}} \sqrt{p(y)} \right) \sqrt{\frac{2 \ln(|\mathcal{Y}|/\delta)}{2n}}$$

where $p(y) = \mathbb{P}(Y = y)$. Substituting this bound into equation 38 with the union bound, we have that with probability at least $1 - \delta$,

$$\mathbb{E}_{X,Y}[\|W_S f_\theta(X) - Y\|] - \frac{1}{n} \sum_{i=1}^{n} \|W_S f_\theta(\bar{x}_i) - \bar{y}_i\| \tag{39}$$

$$\leq \kappa_S \left( \sum_{y \in \tilde{\mathcal{Y}}} \sqrt{\hat{p}(y)} \right) \sqrt{\frac{\ln(2|\tilde{\mathcal{Y}}|/\delta)}{2n}} + \kappa_S \left( \sum_{y \in \mathcal{Y}} \sqrt{p(y)} \right) \sqrt{\frac{2 \ln(2|\mathcal{Y}|/\delta)}{2n}}$$

$$\leq \left( \sum_{y \in \mathcal{Y}} \sqrt{\hat{p}(y)} \right) \kappa_S \sqrt{\frac{2 \ln(2|\mathcal{Y}|/\delta)}{2n}} + \left( \sum_{y \in \mathcal{Y}} \sqrt{p(y)} \right) \kappa_S \sqrt{\frac{2 \ln(2|\mathcal{Y}|/\delta)}{2n}}$$

$$\leq \kappa_S \sqrt{\frac{2 \ln(2|\mathcal{Y}|/\delta)}{2n}} \sum_{y \in \mathcal{Y}} \left( \sqrt{\hat{p}(y)} + \sqrt{p(y)} \right)$$

Combining equation 34 and equation 39 implies that with probability at least $1 - \delta$,

$$\mathbb{E}_{X,Y}[\|W_S f_\theta(X) - Y\|] \tag{40}$$

$$\leq \frac{1}{n} \sum_{i=1}^{n} \|W_S f_\theta(\bar{x}_i) - \bar{y}_i\| + \kappa_S \sqrt{\frac{2 \ln(2|\mathcal{Y}|/\delta)}{2n}} \sum_{y \in \mathcal{Y}} \left( \sqrt{\hat{p}(y)} + \sqrt{p(y)} \right)$$

$$\leq L_S(w_S) + \frac{1}{n} \sum_{i=1}^{n} \|\tilde{W}(f_\theta(\bar{x}_i) - f_\theta(x_i))\|$$

$$+ \frac{1}{n} \sum_{i=1}^{n} \|\varphi(x_i)\| + \frac{1}{n} \sum_{i=1}^{n} \|\varphi(\bar{x}_i)\| + \kappa_S \sqrt{\frac{2 \ln(2|\mathcal{Y}|/\delta)}{2n}} \sum_{y \in \mathcal{Y}} \left( \sqrt{\hat{p}(y)} + \sqrt{p(y)} \right).$$

We will now analyze the term $\frac{1}{n} \sum_{i=1}^{n} \|\varphi(x_i)\| + \frac{1}{n} \sum_{i=1}^{n} \|\varphi(\bar{x}_i)\|$ on the right-hand side of equation 40. Since $W^* = W_{\bar{S}}$,

$$\frac{1}{n} \sum_{i=1}^{n} \|\varphi(x_i)\| = \frac{1}{n} \sum_{i=1}^{n} \|g^*(x_i) - W_{\bar{S}} f_\theta(x_i)\|.$$

By using Hoeffding's inequality (Lemma G.1), we have that for any $\delta > 0$, with probability at least $1 - \delta$,

$$\frac{1}{n} \sum_{i=1}^{n} \|\varphi(x_i)\| \leq \frac{1}{n} \sum_{i=1}^{n} \|g^*(x_i) - W_{\bar{S}} f_\theta(x_i)\| \leq \mathbb{E}_{x^+}[\|g^*(x^+) - W_{\bar{S}} f_\theta(x^+)\|] + \kappa_{\bar{S}} \sqrt{\frac{\ln(1/\delta)}{2n}}.$$

Moreover, by using [50, Theorem 3.1] with the loss function $x^+ \mapsto \|g^*(x^+) - Wf(x^+)\|$ (i.e., Lemma G.2), we have that for any $\delta > 0$, with probability at least $1 - \delta$,

$$\mathbb{E}_{x^+}[\|g^*(x^+) - W_{\bar{S}}f_\theta(x^+)\|] \leq \frac{1}{m}\sum_{i=1}^{m}\|g^*(x_i^+) - W_{\bar{S}}f_\theta(x_i^+)\| + \frac{2\tilde{\mathcal{R}}_m(\mathcal{W} \circ \mathcal{F})}{\sqrt{m}} + \kappa\sqrt{\frac{\ln(1/\delta)}{2m}}$$
(41)

where $\tilde{\mathcal{R}}_m(\mathcal{W} \circ \mathcal{F}) = \frac{1}{\sqrt{m}}\mathbb{E}_{\bar{S},\xi}[\sup_{W \in \mathcal{W}, f \in \mathcal{F}}\sum_{i=1}^{m}\xi_i\|g^*(x_i^+) - Wf(x_i^+)\|]$ is the normalized Rademacher complexity of the set $\{x^+ \mapsto \|g^*(x^+) - Wf(x^+)\| : W \in \mathcal{W}, f \in \mathcal{F}\}$ (it is normalized such that $\tilde{\mathcal{R}}_m(\mathcal{F}) = O(1)$ as $m \to \infty$ for typical choices of $\mathcal{F}$), and $\xi_1, \ldots, \xi_m$ are independent uniform random variables taking values in $\{-1, 1\}$. Takinng union bounds, we have that for any $\delta > 0$, with probability at least $1 - \delta$,

$$\frac{1}{n}\sum_{i=1}^{n}\|\varphi(x_i)\| \leq \frac{1}{m}\sum_{i=1}^{m}\|g^*(x_i^+) - W_{\bar{S}}f_\theta(x_i^+)\| + \frac{2\tilde{\mathcal{R}}_m(\mathcal{W} \circ \mathcal{F})}{\sqrt{m}} + \kappa\sqrt{\frac{\ln(2/\delta)}{2m}} + \kappa_{\bar{S}}\sqrt{\frac{\ln(2/\delta)}{2n}}$$

Similarly, for any $\delta > 0$, with probability at least $1 - \delta$,

$$\frac{1}{n}\sum_{i=1}^{n}\|\varphi(\bar{x}_i)\| \leq \frac{1}{m}\sum_{i=1}^{m}\|g^*(x_i^+) - W_{\bar{S}}f_\theta(x_i^+)\| + \frac{2\tilde{\mathcal{R}}_m(\mathcal{W} \circ \mathcal{F})}{\sqrt{m}} + \kappa\sqrt{\frac{\ln(2/\delta)}{2m}} + \kappa_{\bar{S}}\sqrt{\frac{\ln(2/\delta)}{2n}}.$$

Thus, by taking union bounds, we have that for any $\delta > 0$, with probability at least $1 - \delta$,

$$\frac{1}{n}\sum_{i=1}^{n}\|\varphi(x_i)\| + \frac{1}{n}\sum_{i=1}^{n}\|\varphi(\bar{x}_i)\| \tag{42}$$

$$\leq \frac{2}{m}\sum_{i=1}^{m}\|g^*(x_i^+) - W_{\bar{S}}f_\theta(x_i^+)\| + \frac{4\mathcal{R}_m(\mathcal{W} \circ \mathcal{F})}{\sqrt{m}} + 2\kappa\sqrt{\frac{\ln(4/\delta)}{2m}} + 2\kappa_{\bar{S}}\sqrt{\frac{\ln(4/\delta)}{2n}}$$

To analyze the first term on the right-hand side of equation 42, recall that

$$W_{\bar{S}} = \underset{W'}{\text{minimize}}\|W'\|_F \text{ s.t. } W' \in \underset{W}{\arg\min}\frac{1}{m}\sum_{i=1}^{m}\|Wf_\theta(x_i^+) - g^*(x_i^+)\|^2. \tag{43}$$

Here, since $Wf_\theta(x_i^+) \in \mathbb{R}^r$, we have that

$$Wf_\theta(x_i^+) = \text{vec}[Wf_\theta(x_i^+)] = [f_\theta(x_i^+)^\top \otimes I_r]\,\text{vec}[W] \in \mathbb{R}^r,$$

where $I_r \in \mathbb{R}^{r \times r}$ is the identity matrix, and $[f_\theta(x_i^+)^\top \otimes I_r] \in \mathbb{R}^{r \times dr}$ is the Kronecker product of the two matrices, and $\text{vec}[W] \in \mathbb{R}^{dr}$ is the vectorization of the matrix $W \in \mathbb{R}^{r \times d}$. Thus, by defining $A_i = [f_\theta(x_i^+)^\top \otimes I_r] \in \mathbb{R}^{r \times dr}$ and using the notation of $w = \text{vec}[W]$ and its inverse $W = \text{vec}^{-1}[w]$ (i.e., the inverse of the vectorization from $\mathbb{R}^{r \times d}$ to $\mathbb{R}^{dr}$ with a fixed ordering), we can rewrite equation 43 by

$$W_{\bar{S}} = \text{vec}^{-1}[w_{\bar{S}}] \quad \text{where} \quad w_{\bar{S}} = \underset{w'}{\text{minimize}}\|w'\|_F \text{ s.t. } w' \in \underset{w}{\arg\min}\sum_{i=1}^{m}\|g_i - A_i w\|^2,$$

with $g_i = g^*(x_i^+) \in \mathbb{R}^r$. Since the function $w \mapsto \sum_{i=1}^{m}\|g_i - A_i w\|^2$ is convex, a necessary and sufficient condition of the minimizer of this function is obtained by

$$0 = \nabla_w \sum_{i=1}^{m}\|g_i - A_i w\|^2 = 2\sum_{i=1}^{m}A_i^\top(g_i - A_i w) \in \mathbb{R}^{dr}$$

This implies that

$$\sum_{i=1}^{m}A_i^\top A_i w = \sum_{i=1}^{m}A_i^\top g_i.$$

In other words,

$$A^\top A w = A^\top g \quad \text{where } A = \begin{bmatrix} A_1 \\ A_2 \\ \vdots \\ A_m \end{bmatrix} \in \mathbb{R}^{mr \times dr} \text{ and } g = \begin{bmatrix} g_1 \\ g_2 \\ \vdots \\ g_m \end{bmatrix} \in \mathbb{R}^{mr}$$

Thus,

$$w' \in \arg\min_w \sum_{i=1}^m \|g_i - A_i w\|^2 = \{(A^\top A)^\dagger A^\top g + v : v \in \text{Null}(A)\}$$

where $(A^\top A)^\dagger$ is the Moore–Penrose inverse of the matrix $A^\top A$ and $\text{Null}(A)$ is the null space of the matrix $A$. Thus, the minimum norm solution is obtained by

$$\text{vec}[W_{\bar{S}}] = w_{\bar{S}} = (A^\top A)^\dagger A^\top g.$$

Thus, by using this $W_{\bar{S}}$, we have that

$$\frac{1}{m}\sum_{i=1}^m \|g^*(x_i^+) - W_{\bar{S}} f_\theta(x_i^+)\| = \frac{1}{m}\sum_{i=1}^m \sqrt{\sum_{k=1}^r ((g_i - A_i w_{\bar{S}})_k)^2}$$

$$\leq \sqrt{\frac{1}{m}\sum_{i=1}^m \sum_{k=1}^r ((g_i - A_i w_{\bar{S}})_k)^2}$$

$$= \frac{1}{\sqrt{m}}\sqrt{\sum_{i=1}^m \sum_{k=1}^r ((g_i - A_i w_{\bar{S}})_k)^2}$$

$$= \frac{1}{\sqrt{m}}\|g - A w_{\bar{S}}\|_2$$

$$= \frac{1}{\sqrt{m}}\|g - A(A^\top A)^\dagger A^\top g\|_2 = \frac{1}{\sqrt{m}}\|(I - A(A^\top A)^\dagger A^\top)g\|_2$$

where the inequality follows from the Jensen's inequality and the concavity of the square root function. Thus, we have that

$$\frac{1}{n}\sum_{i=1}^n \|\varphi(x_i)\| + \frac{1}{n}\sum_{i=1}^n \|\varphi(\bar{x}_i)\| \tag{44}$$

$$\leq \frac{2}{\sqrt{m}}\|(I - A(A^\top A)^\dagger A^\top)g\|_2 + \frac{4\mathcal{R}_m(\mathcal{W} \circ \mathcal{F})}{\sqrt{m}} + 2\kappa\sqrt{\frac{\ln(4/\delta)}{2m}} + 2\kappa_{\bar{S}}\sqrt{\frac{\ln(4/\delta)}{2n}}$$

By combining equation 40 and equation 44 with union bound, we have that

$$\mathbb{E}_{X,Y}[\|W_S f_\theta(X) - Y\|] \tag{45}$$

$$\leq L_S(w_S) + \frac{1}{n}\sum_{i=1}^n \|\tilde{W}(f_\theta(\bar{x}_i) - f_\theta(x_i))\| + \frac{2}{\sqrt{m}}\|\mathbf{P}_A g\|_2$$

$$+ \frac{4\mathcal{R}_m(\mathcal{W} \circ \mathcal{F})}{\sqrt{m}} + 2\kappa\sqrt{\frac{\ln(8/\delta)}{2m}} + 2\kappa_{\bar{S}}\sqrt{\frac{\ln(8/\delta)}{2n}}$$

$$+ \kappa_S\sqrt{\frac{2\ln(4|\mathcal{Y}|/\delta)}{2n}}\sum_{y \in \mathcal{Y}}\left(\sqrt{\hat{p}(y)} + \sqrt{p(y)}\right).$$

where $\tilde{W} = W_S - W^*$ and $\mathbf{P}_A = I - A(A^\top A)^\dagger A^\top$.

We will now analyze the second term on the right-hand side of equation 45:

$$\frac{1}{n}\sum_{i=1}^n \|\tilde{W}(f_\theta(\bar{x}_i) - f_\theta(x_i))\| \leq \|\tilde{W}\|_2 \left(\frac{1}{n}\sum_{i=1}^n \|f_\theta(\bar{x}_i) - f_\theta(x_i)\|\right), \tag{46}$$

where $\|\tilde{W}\|_2$ is the spectral norm of $\tilde{W}$. Since $\bar{x}_i$ shares the same label with $x_i$ as $\bar{x}_i \sim \mathcal{D}_{y_i}$ (and $x_i \sim \mathcal{D}_{y_i}$), and because $f_\theta$ is trained with the unlabeled data $\bar{S}$, using Hoeffding's inequality (Lemma G.1) implies that with probability at least $1 - \delta$,

$$\frac{1}{n}\sum_{i=1}^{n}\|f_\theta(\bar{x}_i) - f_\theta(x_i)\| \le \mathbb{E}_{y\sim\rho}\mathbb{E}_{\bar{x},x\sim\mathcal{D}_y^2}[\|f_\theta(\bar{x}) - f_\theta(x)\|] + \tau_{\bar{S}}\sqrt{\frac{\ln(1/\delta)}{2n}}. \tag{47}$$

Moreover, by using [50, Theorem 3.1] with the loss function $(x, \bar{x}) \mapsto \|f_\theta(\bar{x}) - f_\theta(x)\|$ (i.e., Lemma G.2), we have that with probability at least $1 - \delta$,

$$\mathbb{E}_{y\sim\rho}\mathbb{E}_{\bar{x},x\sim\mathcal{D}_y^2}[\|f_\theta(\bar{x}) - f_\theta(x)\|] \le \frac{1}{m}\sum_{i=1}^{m}\|f_\theta(x_i^+) - f_\theta(x_i^{++})\| + \frac{2\tilde{\mathcal{R}}_m(\mathcal{F})}{\sqrt{m}} + \tau\sqrt{\frac{\ln(1/\delta)}{2m}} \tag{48}$$

where $\tilde{\mathcal{R}}_m(\mathcal{F}) = \frac{1}{\sqrt{m}}\mathbb{E}_{\bar{S},\xi}[\sup_{f\in\mathcal{F}}\sum_{i=1}^{m}\xi_i\|f(x_i^+) - f(x_i^{++})\|]$ is the normalized Rademacher complexity of the set $\{(x^+, x^{++}) \mapsto \|f(x^+) - f(x^{++})\| : f \in \mathcal{F}\}$ (it is normalized such that $\tilde{\mathcal{R}}_m(\mathcal{F}) = O(1)$ as $m \to \infty$ for typical choices of $\mathcal{F}$), and $\xi_1, \ldots, \xi_m$ are independent uniform random variables taking values in $\{-1, 1\}$. Thus, taking union bound, we have that for any $\delta > 0$, with probability at least $1 - \delta$,

$$\frac{1}{n}\sum_{i=1}^{n}\|\tilde{W}(f_\theta(\bar{x}_i) - f_\theta(x_i))\| \tag{49}$$

$$\le \|\tilde{W}\|_2\left(\frac{1}{m}\sum_{i=1}^{m}\|f_\theta(x_i^+) - f_\theta(x_i^{++})\| + \frac{2\tilde{\mathcal{R}}_m(\mathcal{F})}{\sqrt{m}} + \tau\sqrt{\frac{\ln(2/\delta)}{2m}} + +\tau_{\bar{S}}\sqrt{\frac{\ln(2/\delta)}{2n}}\right).$$

By combining equation 45 and equation 49 using the union bound, we have that with probability at least $1 - \delta$,

$$\mathbb{E}_{X,Y}[\|W_S f_\theta(X) - Y\|] \tag{50}$$

$$\le L_S(w_S) + \|\tilde{W}\|_2\left(\frac{1}{m}\sum_{i=1}^{m}\|f_\theta(x_i^+) - f_\theta(x_i^{++})\| + \frac{2\tilde{\mathcal{R}}_m(\mathcal{F})}{\sqrt{m}} + \tau\sqrt{\frac{\ln(4/\delta)}{2m}} + \tau_{\bar{S}}\sqrt{\frac{\ln(4/\delta)}{2n}}\right)$$

$$+ \frac{2}{\sqrt{m}}\|\mathbf{P}_A g\|_2 + \frac{4\mathcal{R}_m(\mathcal{W}\circ\mathcal{F})}{\sqrt{m}} + 2\kappa\sqrt{\frac{\ln(16/\delta)}{2m}} + 2\kappa_{\bar{S}}\sqrt{\frac{\ln(16/\delta)}{2n}}$$

$$+ \kappa_S\sqrt{\frac{2\ln(8|\mathcal{Y}|/\delta)}{2n}}\sum_{y\in\mathcal{Y}}\left(\sqrt{\hat{p}(y)} + \sqrt{p(y)}\right)$$

$$= L_S(w_S) + \|\tilde{W}\|_2\left(\frac{1}{m}\sum_{i=1}^{m}\|f_\theta(x_i^+) - f_\theta(x_i^{++})\|\right) + \frac{2}{\sqrt{m}}\|\mathbf{P}_A g\|_2 + Q_{m,n}$$

where

$$Q_{m,n} = \|\tilde{W}\|_2\left(\frac{2\tilde{\mathcal{R}}_m(\mathcal{F})}{\sqrt{m}} + \tau\sqrt{\frac{\ln(3/\delta)}{2m}} + \tau_{\bar{S}}\sqrt{\frac{\ln(3/\delta)}{2n}}\right)$$

$$+ \kappa_S\sqrt{\frac{2\ln(6|\mathcal{Y}|/\delta)}{2n}}\sum_{y\in\mathcal{Y}}\left(\sqrt{\hat{p}(y)} + \sqrt{p(y)}\right)$$

$$+ \frac{4\mathcal{R}_m(\mathcal{W}\circ\mathcal{F})}{\sqrt{m}} + 2\kappa\sqrt{\frac{\ln(4/\delta)}{2m}} + 2\kappa_{\bar{S}}\sqrt{\frac{\ln(4/\delta)}{2n}}.$$

Define $Z_{\bar{S}} = [f(x_1^+), \ldots, f(x_m^+)] \in \mathbb{R}^{d\times m}$. Then, we have $A = [Z_{\bar{S}}^\top \otimes I_r]$. Thus,

$$\mathbf{P}_A = I - [Z_{\bar{S}}^\top \otimes I_r][Z_{\bar{S}}Z_{\bar{S}}^\top \otimes I_r]^\dagger[Z_{\bar{S}} \otimes I_r] = I - [Z_{\bar{S}}^\top(Z_{\bar{S}}Z_{\bar{S}}^\top)^\dagger Z_{\bar{S}} \otimes I_r] = [\mathbf{P}_{Z_{\bar{S}}} \otimes I_r]$$

where $\mathbf{P}_{Z_{\bar{S}}} = I_m - Z_{\bar{S}}^\top(Z_{\bar{S}}Z_{\bar{S}}^\top)^\dagger Z_{\bar{S}} \in \mathbb{R}^{m\times m}$. By defining $Y_{\bar{S}} = [g^*(x_1^+), \ldots, g^*(x_m^+)]^\top \in \mathbb{R}^{m\times r}$, since $g = \text{vec}[Y_{\bar{S}}^\top]$,

$$\|\mathbf{P}_A g\|_2 = \|[\mathbf{P}_{Z_{\bar{S}}} \otimes I_r]\text{vec}[Y_{\bar{S}}^\top]\|_2 = \|\text{vec}[Y_{\bar{S}}^\top\mathbf{P}_{Z_{\bar{S}}}]\|_2 = \|\mathbf{P}_{Z_{\bar{S}}}Y_{\bar{S}}\|_F \tag{51}$$

On the other hand, recall that $W_S$ is the minimum norm solution as

$$W_S = \underset{W'}{\text{minimize}} \ \|W'\|_F \ \text{s.t.} \ W' \in \arg\min_W \frac{1}{n} \sum_{i=1}^{n} \|W f_\theta(x_i) - y_i\|^2.$$

By solving this, we have

$$W_S = Y^\top Z_S{}^\top (Z_S Z_S{}^\top)^\dagger,$$

where $Z_S = [f(x_1), \ldots, f(x_n)] \in \mathbb{R}^{d \times n}$ and $Y_S = [y_1, \ldots, y_n]^\top \in \mathbb{R}^{n \times r}$. Then,

$$L_S(w_S) = \frac{1}{n} \sum_{i=1}^{n} \|W_S f_\theta(x_i) - y_i\| = \frac{1}{n} \sum_{i=1}^{n} \sqrt{\sum_{k=1}^{r} ((W_S f_\theta(x_i) - y_i)_k)^2}$$

$$\leq \sqrt{\frac{1}{n} \sum_{i=1}^{n} \sum_{k=1}^{r} ((W_S f_\theta(x_i) - y_i)_k)^2}$$

$$= \frac{1}{\sqrt{n}} \|W_S Z_S - Y^\top\|_F$$

$$= \frac{1}{\sqrt{n}} \|Y^\top (Z_S{}^\top (Z_S Z_S{}^\top)^\dagger Z_S - I)\|_F$$

$$= \frac{1}{\sqrt{n}} \|(I - Z_S{}^\top (Z_S Z_S{}^\top)^\dagger Z_S) Y\|_F$$

Thus,

$$L_S(w_S) = \frac{1}{\sqrt{n}} \|\mathbf{P}_{Z_S} Y\|_F \tag{52}$$

where $\mathbf{P}_{Z_S} = I - Z_S{}^\top (Z_S Z_S{}^\top)^\dagger Z_S$.

By combining equation 50–equation 52 and using $1 \leq \sqrt{2}$, we have that with probability at least $1 - \delta$,

$$\mathbb{E}_{X,Y}[\|W_S f_\theta(X) - Y\|] \leq c I_{\bar{S}}(f_\theta) + \frac{2}{\sqrt{m}} \|\mathbf{P}_{Z_{\bar{S}}} Y_{\bar{S}}\|_F + \frac{1}{\sqrt{n}} \|\mathbf{P}_{Z_S} Y_S\|_F + Q_{m,n}, \tag{53}$$

where

$$Q_{m,n} = c \left( \frac{2\tilde{\mathcal{R}}_m(\mathcal{F})}{\sqrt{m}} + \tau \sqrt{\frac{\ln(3/\delta)}{2m}} + \tau_{\bar{S}} \sqrt{\frac{\ln(3/\delta)}{2n}} \right)$$

$$+ \kappa_S \sqrt{\frac{2 \ln(6|\mathcal{Y}|/\delta)}{2n}} \sum_{y \in \mathcal{Y}} \left( \sqrt{\hat{p}(y)} + \sqrt{p(y)} \right)$$

$$+ \frac{4\mathcal{R}_m(\mathcal{W} \circ \mathcal{F})}{\sqrt{m}} + 2\kappa \sqrt{\frac{\ln(4/\delta)}{2m}} + 2\kappa_{\bar{S}} \sqrt{\frac{\ln(4/\delta)}{2n}}.$$

$\square$

Now,

# Appendix J    Information Optimization and the VICReg Objective

*Assumption* 1. The eigenvalues of $\Sigma(x_j)$ are in some range $a \leq \lambda(\Sigma(x_j)) \leq b$.

*Assumption* 2. The differences between the means of the Gaussians are bounded

$$M = \max_{i,j} \|\mu(X_i) - \mu(X_j)\|^2$$

**Lemma J.1.** *The maximum eigenvalue of each $\mu(X_j)\mu(X_j)^T$ is at most $M$.*

*Proof.* The term $\mu(X_j)\mu(X_j)^T$ is an outer product of the mean vector $\mu(X_j)$, which is a symmetric matrix. The eigenvalues of a symmetric matrix are equal to the squares of the singular values of the original matrix. Since the singular values of a vector are equal to its absolute values, the maximum eigenvalue of $\mu(X_j)\mu(X_j)^T$ is equal to the square of the maximum absolute value of $\mu(X_j)$. By the second assumption, this is at most $M$. $\qquad\square$

**Lemma J.2.** *The maximum eigenvalue of $-\mu_Z\mu_Z^T$ is non-positive and its absolute value is at most $M$.*

*Proof.* The term $-\mu_Z\mu_Z^T$ is a negative outer product of the overall mean vector $\mu_Z$, which is a symmetric matrix. Its eigenvalues are non-positive and equal to the negative squares of the singular values of $\mu_Z$. Since the singular values of a vector are equal to its absolute values, the absolute value of the maximum eigenvalue of $-\mu_Z\mu_Z^T$ is equal to the square of the maximum absolute value of $\mu_Z$, which is also bounded by $M$ by the second assumption. $\qquad\square$

**Lemma J.3.** *The sum of the eigenvalues of $\Sigma_Z$ is bounded*

$$\sum_i \lambda_i(Z) \leq (b + M) \times K$$

*Proof.* Given a Gaussian mixture model where each component $Z|x_j$ has mean $\mu(X_j)$ and covariance matrix $\Sigma(x_j)$, the mixture can be written as:

$$Z = \sum_j p_j Z|x_j$$

where $p_j$ are the mixing coefficients. The covariance matrix of the mixture, $\Sigma_Z$, is then given by:

$$\Sigma_Z = \sum_j p_j \left(\Sigma(x_j) + \mu(X_j)\mu(X_j)^T\right) - \mu_Z\mu_Z^T$$

where $\mu_Z$ is the mean of the mixture distribution.

By Lemmas 1.1, 1.2, and assumptions 1 and 2, the maximum eigenvalues of $(\Sigma(x_j), \mu(X_j)\mu(X_j)^T$ and $\mu_Z\mu_Z^T$. are at most $b$, $M$, and $M$, respectively. Therefore, by Weyl's inequality for the sum of two symmetric matrices, the maximum eigenvalue of $\Sigma_Z$ is at most $b + M$.

$$\lambda_{max}(\Sigma_Z) \leq \frac{1}{K}\sum_{i=1}^{K}(max(\lambda(\Sigma(X_i))) + M) \leq b + M$$

It means that we can bound the sum of the eigenvalues of $\Sigma_Z$ with

$$\sum_i \lambda_i(\Sigma_Z) \leq (b + M) \times K$$

$\qquad\square$

**Lemma J.4.** *Let $\Sigma_Z$ be a positive semidefinite matrix of size $N \times N$. Consider the optimization problem given by:*

$$maximize \ \log\det(\Sigma_Z)$$

$$such \ that:$$

$$\sum_{i=1}^{N} \lambda_i(\Sigma_Z) \leq c$$

$$\Sigma_Z \succeq 0$$

*where $\lambda_i(\Sigma_Z)$ denotes the $i$-th eigenvalue of $\Sigma_Z$ and $c$ is a constant. The solution to this problem is a diagonal matrix with equal diagonal elements.*

*Proof.* The determinant of a matrix is the product of its eigenvalues, so the objective function $\log \det(\Sigma_Z)$ can be rewritten as $\sum_{i=1}^{N} \log(\lambda_i(\Sigma_Z))$. Our problem is then to maximize this sum under the constraints that the sum of the eigenvalues does not exceed $c$ and that $\Sigma_Z$ is positive semi-definite.

Applying Jensen's inequality to the concave function $\log(x)$ with weights $1/N$, we find that $\frac{1}{N} \sum_{i=1}^{N} \log(\lambda_i(\Sigma_Z)) \leq \log(\frac{1}{N} \sum_{i=1}^{N} \lambda_i(\Sigma_Z))$. Equality holds if and only if all $\lambda_i(\Sigma_Z)$ are equal.

Setting $\lambda_i(\Sigma_Z) = x$ for all $i$, we see that the constraint $\sum_{i=1}^{N} \lambda_i(\Sigma_Z) \leq c$ becomes $Nx \leq c$, leading to the optimal eigenvalue $x = c/N$ under the constraint.

Since $\Sigma_Z$ is positive semi-definite, it can be diagonalized via an orthogonal transformation without changing the sum of its eigenvalues or its determinant. Therefore, the solution to the problem is a diagonal matrix with all diagonal entries equal to $c/N$.

This completes the proof. $\qquad\square$

**Theorem J.5.** *Given a Gaussian mixture model where each component $Z|X_i$ has covariance matrix $\Sigma(X_i)$, under the assumptions above, the solution to the optimization problem*

$$\text{maximize} \sum_i \log \frac{|\Sigma_Z|}{|\Sigma(X_i)|}$$

*is a diagonal matrix $\Sigma_Z$ with equal diagonal elements.*

*Proof.* The objective function can be decomposed as follows:

$$\sum_i \log \frac{|\Sigma_Z|}{|\Sigma(X_i)|} = \sum_i \left( \log |\Sigma_Z| - \log |\Sigma(X_i)| \right)$$
$$= K \log |\Sigma_Z| - \sum_i \log |\Sigma(X_i)| ,$$

where $K$ is the number of components in the Gaussian mixture model.

In this optimization problem, we are optimizing over $\Sigma_Z$. The term $\sum_i \log |\Sigma(X_i)|$ is constant with respect to $\Sigma_Z$, therefore we can focus on maximizing $K \log |\Sigma_Z|$.

As the determinant of a matrix is the product of its eigenvalues, $\log |\Sigma_Z|$ is the sum of the logs of the eigenvalues of $\Sigma_Z$. Thus, maximizing $\log |\Sigma_Z|$ corresponds to maximizing the sum of the logarithms of the eigenvalues of $\Sigma_Z$.

According to Lemma 1.4, when we have a constraint on the sum of the eigenvalues, the solution to the problem of maximizing the sum of the logarithms of the eigenvalues of a positive semidefinite matrix $\Sigma_Z$ is a diagonal matrix with equal diagonal elements.

From Lemma 1.3, we know that the sum of the eigenvalues of $\Sigma_Z$ is bounded by $(b + M) \times K$. Therefore, when we maximize $K \log |\Sigma_Z|$ under these constraints, the solution will be a diagonal matrix with equal diagonal elements. This completes the proof of the theorem. $\qquad\square$

## Appendix K    Entropy Comparison - Experimental Details

We use ResNet-18 [32] as our backbone. Each model is trained with $512$ batch size for $800$ epochs. We use the SGD optimizer with a momentum of $0.9$ and a weight decay of $1e^{-4}$. The initial learning rate is $0.5$. This learning rate follows the cosine decay with a linear warmup schedule. For augmentation, two augmented versions of each input image are generated. During this process, each image is cropped with random size, and resized to the original resolution, followed by random applications of horizontal mirroring, color jittering, grayscale conversion, Gaussian blurring and solarization. For the entropy estimation, we use the same method as in [38], which uses a lower bound of the entropy using the distances of the representations under the assumption of a mixture of the Gaussians around the representations with constant variance.

$$H(Z) := H(Z|C) - \sum_i c_i \ln \sum_j c_j e^{-D(p_i||p_j)}. \tag{54}$$

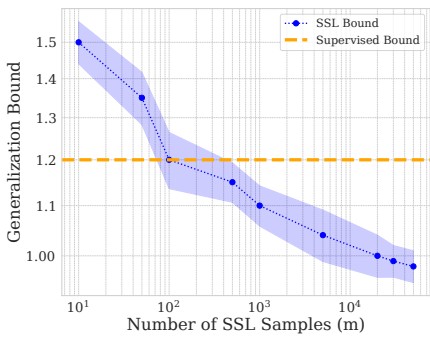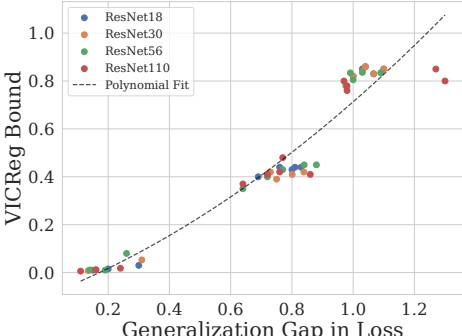

Figure 5: **Our generalization bound predicts more accurately the generalization gap in the loss. (left)** Our SSL VICReg generalization bound outperforms state-of-the-art supervised generalization bounds. **(right)** Strong correlation between the generalization gap and our generalization bound for VICReg. Pearson correlation - 0.9633. Conducted on CIFAR-10.

where $p_i$ and $p_j$ are the distributions of the representation of the i-th and j-th examples, and $D(p_i||p_j)$ represents the divergence between these distributions. Also, $c_i$ indicates the weight of component $i$ ($c_i \geq 0$, $\sum_i c_i = 1$), and $C$ is a discrete random variable where $P(C = i) = c_i$ .

## Appendix L   Reproducibility Statement

All of the methods in our study are based on existing methods and their open-source implementations. We provide a detailed implementation setup for both the pre-training and downstream experiments. Additionally, we are committed to ensuring reproducibility and open science. Therefore, after publication, we will provide pretrained checkpoints and make the code openly available on a public repository. This will enable other researchers to reproduce our results and build upon our work.

## Appendix M   Expriemtns on the generalization bound

In this section we have more empirical results on the connection between our generalization to the generalization gap.

## Appendix N   Limitations of assuming inherent network randomness

First, we provide a brief overview of the challenges associated with estimating mutual information (MI) in DNNs. This context will set the stage for a detailed discussion on the advantages of our method

Estimating MI in DNNs poses various practical challenges. For example, deterministic DNNs with strict monotonic nonlinearities yield infinite or constant MI, complicating optimization [4] or making optimization ill-posed. One solution is to assume an unknown distribution over the representation and estimating MI directly [9, 57]. However, this is challenging theoretically and requires large sample sizes [55, 47] and is prone to significant estimation errors [70].

To address these issues, prior works have proposed various assumptions about randomness in neural networks. For example, [1] posits noise in DNNs' parameters, while [1] assumes noise in the representations. These assumptions are valid if they outperform deterministic DNNs or yield similar representations to those trained in practice.

Our empirical evidence suggests that noise in the input has much better performance and representations similar to the deterministic network compared to noise in the network itself. We compared our model, which introduces noise at the input level, with models that add noise to network representations [26], using CIFAR-100 and Tiny-ImageNet and VICReg with ConvNeXt. The results,

tabulated below, indicate that the noise injected into the representation has much worse performance degradation compared to our input noise model.

Table 2: Comparison of Noisy Network and Noisy Input across different datasets for different noise levels

| $\beta$ (Noise Level) | Tiny-ImageNet | | CIFAR100 | |
|---|---|---|---|---|
| | Noisy Network | Noisy Input (ours) | Noisy Network | Noisy Input (ours) |
| $\beta = 0$ (no noise) | 53.1 | 53.1 | 70.1 | 70.1 |
| $\beta = 0.05$ | 51.7 | 53.0 | 69.7 | 70.0 |
| $\beta = 0.1$ | 50.2 | 52.8 | 68.8 | 69.6 |
| $\beta = 0.2$ | 48.1 | 52.3 | 67.1 | 68.9 |

Our approach, focusing on input noise, does not necessitate explicit noise injection during training—unlike methods assuming inherent network noise.

Lastly, we validate that the assumption of input noise leads to better alignment with deterministic networks. Following [14], we used Centered Kernel Alignment (CKA) [18] to compare deterministic DNNs with our noise model (input noise) to DNNs with noise in the representations [26]. The results, also tabulated below, confirm that assuming input noise aligns more closely with deterministic DNN representations than assuming noise in the DNN parameters.

Table 3: Performance comparison of different methods at varying noise levels

| Noise Level/Method | Deterministic Network | Noisy Network | Noisy Input (our method) |
|---|---|---|---|
| $\beta = 0.05$ | 0.97 | 0.82 | 0.93 |
| $\beta = 0.1$ | 0.97 | 0.69 | 0.85 |
| $\beta = 0.2$ | 0.97 | 0.54 | 0.77 |
| $\beta = 0.3$ | 0.97 | 0.32 | 0.69 |

In summary, our input noise model more effectively captures deterministic DNN behavior and is more robust than assuming inherent network randomness.

## Appendix O  Boader Impact

In the landscape of machine learning, SSL algorithms have found a broad array of practical applications, ranging from image recognition to natural language processing. With the proliferation of data in recent years, their utility has grown exponentially, often serving as key components in numerous real-world use cases. This paper introduces a new family of SSL algorithms that have demonstrated superior performance. Given the widespread use of SSL algorithms, the advancements proposed in this paper have the potential to considerably enhance the effectiveness of systems that rely on these techniques. As a result, the impact of these improved algorithms could be far-reaching, potentially influencing a multitude of applications and sectors where machine learning is currently employed. With this potential for value also comes the potential that proposed methods make false promises that will not benefit real-world practitioners and may, in fact, cause harm when deployed in sensitive applications. For this reason, we release our numerical results and implementation details for the sake of transparency and reproducibility. As with all new state-of-the-art methods, our improvements may also improve models used for malicious intentions.

