# OpenReview forum: "An Information Theory Perspective on Variance-Invariance-Covariance Regularization"
_NeurIPS.cc/2023/Conference — NeurIPS 2023 poster_

### Official Review · Reviewer_2fL7 · 2023-06-30

**Soundness:** 3 good
**Presentation:** 2 fair
**Contribution:** 2 fair
**Rating:** 5
**Confidence:** 3

**Summary:**

This paper presents an information-theoretic analysis of the self supervised learning algorithm VICReg. The authors first introduce an approach to study neural networks using an information-theoretic perspective by shifting the stochasticity from the networks to the input data. Therefore they derive an information-theoretic objective (equation 7) that is linked (through a series of approximations) to the VICReg objective. The assumptions of the proposed theory are validated empirically using ResNet-18 across various datasets.

The paper also proposes a plug-in entropy estimator that can be merged with the objective term of other SSL methods for entropy optimization. The authors shows empirically that doing so with VICReg and training a ResNet-18 architecture on CIFAR-10 dataset leads to an improvement on the test accuracy compared to the original VICReg method.

Lastly the paper derives a generalization bound for VICReg that provides a probabilistic guarantee on the downstream generalization ability of this SSL method.

**Strengths:**

- The perspective offered by the new analysis is interesting and could have practical implications.


**Weaknesses:**

- I find some of the explanation (example Section 4) is not immediately clear. There are some statements that should be verified and could be clarified in the manuscript in order to make the paper more accessible (see questions below on clarifications).
- I find the empirical analysis of the practical implications weak: experiments are limited to CIFAR 10 on ResNet18. I believe there could be an opportunity to strengthen the paper by providing a more thorough evaluation on larger architectures and more challenging datasets (see questions below on experimental evaluation for some suggestions).
- The claim that the proposed plug-in entropy estimator could help other SSL methods (beyond VicReg) is not supported by empirical evaluation.
- The paper contains some significant errors: punctuation missing, use of different styles to refer to the appendix, missing figure citations, a few grammar errors... See Minor Notes below for a non-exhaustive list.

**Minor Notes**
- Line 156: a full stop seems to be missing “of prototypes N As”.
- Line 227: missing full stop at the end of the page.
- Line 242: missing closed parenthesis.
- Notation is not consistent, Appendix A, B, C... vs appendix 1,2,3....
- Figure 2 is not referenced in the text.
- Table 1 missing closed parenthesis on the last line.
- Line 314 Grammar: “label classes to go infinity”.
- Line 347: Grammar: “By transferring the required stochasticity required “ .

**Questions:**

As mentioned above the biggest weaknesses are related to some theoretical parts that are not clear, and the experimental evaluation which is not sufficiently convincing. Here are some opportunities to make the paper stronger (and potentially the rating higher).

**Clarifications**
Providing more information here could help improve the Soundness and Presentation rating.

- In Equation 4, could the authors clarify how the inequality holds? Even in Appendix A this first step is taken for granted but it might be helpful to explain it, at least in the Appendix.
- In Section 4.1 it is not clear to me what happened to the invariance term when going from equation 7 to equation 8. That term seems to have been dropped. Could the authors clarify this?
- “We sampled 512 Gaussian samples for each image” I suppose this means that given 1 image noise is added by sampling 512 different noise that is added to the image? Could this be clarified?
- “However, these methods (reviewer: these methods being [61, 65]) assume that randomness comes from the DNN, contrary to common practice.” Could the author clarify where in 61 and 65 this is assumed?
- “the mutual information between the input and the DNN representation is infinite”. This statement is presented as always true but I do not understand how this would be always guaranteed. None of the papers cited in this section ([8,17,28,37,52,61,…]) seem to support this statement. Do the authors refer to the difficulties of computing MI in continuous domain or could the authors clarify this statement?
- It seems that the loss function of VICReg reported in this paper has some inaccuracies. For example, there is a \gamma parameter that appears twice (in the regularization and the invariance term) but odes not match the loss in [5]. Since in [5] \gamma was fixed to 1 it might not make much difference but it would be better to report the correct equation.
- In section 5.1 “The results showed that entropy decreased for all methods during training, with SimCLR having the lowest and VICReg the highest entropy.” From Figure 2 it seems the entropy is increasing not decreasing. Could the authors clarify?
- “The entire input–output mapping then becomes a CPA spline “. I wonder if this statement is correct. According to [4] DNNs can be seen as a composition of max-affine spline operators, which I am not sure is the same as stated in the paper under review. Could the authors provide more information?
- How many trial were used to compute the error bars in Table 1?
- “the invariance term of current SSL methods with plug-in methods for optimizing the entropy.” Is it merged with the invariance or with the regularizer term?

**Experimental Evaluation**
- The downstream task results are only presented on CIFAR-10 and ResNet-18. Unfortunately, in my experience not everything that works on CIFAR-10 (and on a small architecture) transfers to more realistic datasets (ImageNet would be ideal, TinyImageNet or CIFAR-100 would at least be better than CIFAR-10). The paper would be stronger with these experiments.
- Since the authors claim that the plug-in estimators could be beneficial for “other SSL methods“ it would be great to show how it affects other SSL methods beyond VICReg.
- Other works, e.g., [A], have found that entropy maximization also leads to other beneficial aspects beyond downstream task performance. Specifically, it seems to lead to easier and more robust training with respect to smaller batch size or different EMA parameters (where EMA is used). A similar analysis would also improve and strengthen the practical implication of this work.
- Section 5.1 seems only marginally related to the rest of the paper. However, it would be interesting (and would connect more tightly this section to the rest of the paper) to show in Figure 2 the evolution of the entropy of VICReg (and other algorithms, see points below) with the proposed entropy plug-in estimator to show that indeed it leads to higher entropy.

[A] “The Role of Entropy and Reconstruction in Multi-View Self-Supervised Learning” Galvez et al. ICML2023




**Limitations:**

Limitations are not clearly described in the main paper. They are present in the appendix but I believe they would be more useful in the main part of the manuscript. Additionally in the limitation section of the appendix the authors state “Furthermore, our study hinges on certain assumptions that, despite rigorous validation efforts, may not hold universally” which is fair but it would be even better and more precise to clarify which are such assumptions.

---

> ### Author Rebuttal · Authors · 2023-08-09
>
> Dear Reviewer 2fL7,
>
> We appreciate the comprehensive review and your valuable insights. Your insights will significantly enhance our paper. We want to address your comments in detail:
>
>
> ## Extending the Empirical Evaluations
>
> Our paper emphasizes VICReg's theoretical aspects and advantages over other methods. Nevertheless, we acknowledge that additional empirical studies could amplify our findings. Therefore, we have enriched our experiments by:
>
> - **Exploring Larger Datasets**: We executed our models on the
> Tiny-ImageNet [1] and CIFAR-100 datasets.
> - **Integrating Recent Architectures**: We incorporated modern architectures, including the Vision Transformer (VIT) [2] and ConvNetX [3].
> - **Incorporating Additional SSL Methods**: Our analysis was expanded to encompass methods like MoCo[4] and SwAV[5].
>
> Our updated results (Table 1 in the attached page) underline substantial enhancements of our plug-in estimators across diverse methods, reaffirming our theoretical assertions and underscoring the adaptability of our approach.
>
> In agreement with your suggestion, we have included **the evolution of entropy using our proposed estimator** in the attached page,  which shows increased entropy that enables better downstream task classification.
>
>
> ## Minor Notes:
> We will fix these suggestions in our revised version.
>
> ## Clarifications
> 1. **Equation 4:**  We agree that the inequality warrants an explicit explanation. The inequality arises from the decomposition of p(z|x'), where the expectation is over p(X'), to p(z|z')p(z'|x'), where we first sample from p(X') and then sample from p(Z'|x').
> 2. **Invariance term in Equations 7 and 8:** This term is included in Equation 8 because it is also a component of the original VICReg objective. Equation 8 is not equal to Equation 7; it is only part of the objective we optimize. In Section 4.1, we aimed to analyze the remaining terms and compare them to the VICReg objective, given that the invariance term is the same in both objectives.
> 3. **Gaussian samples for each image:** You are correct.
> 4. **Randomness in DNNs:** We apologize for the typo. "However, these methods..." should precede the sentence "Other research has presumed...". The methods that assume randomness originates from the DNN are referenced as [17], [37], and [52], considering a deterministic input and a variational DNN.
> 5. **Infinite Mutual Information (MI)**. This concept was explored in depth in [1]. Two main issues arise when discussing MI in DNNs:
> (i) Estimating MI directly can be challenging, even with an underlying distribution [2,3].
> (ii) MI can often be infinite or a piecewise constant function of the network's parameters. Theorem 1 in [1] showed that MI is infinite for continuous inputs with almost all non-trivial parameter choices, and it becomes a piecewise constant function for discrete inputs, complicating optimization.
>
> 6. **VICReg loss function:** We apologize for the confusion. The \gamma in the variance loss differs from the invariance term's coefficient.
>
> 7. **Entropy in Figure 2:** We apologize for the typo. The entropy in Figure 2 is indeed increasing, not decreasing, during training, as our analysis suggested.
>
> 8. **DNNs as CPA splines:**  Lets first consider a single layer mapping made of an affine transformation followed by a nonlinearity such as ReLU or Leaky-ReLU. That nonlinearity is a continuous piecewise affine mapping (a spline of order 1). Taking ReLU for illustration, the mapping is 0 from - infinity to 0, and then 1 from 0 to infinity, making the mapping continuous and affine on each of its two regions ((-infinity,0], (0, infinity)). Composing that affine spline (activation) with the preceding affine mapping preserves that property, i.e., the entire layer is an affine spline. Max-affine splines (as named in [8]) are simply a particular type of affine splines where the activation can be expressed using a pointwise maximum (recall that ReLU(x) = max(0,x)), that naming, however, does not impact the fact that a layer is in fact an affine spline. Moving to the entire input-output mapping of a DNN, i.e., a composition of layers, we can use the previous statement. We know that a DNN is a composition of affine splines. As such, the entire mapping is itself an affine spline. Detailed derivations are provided in [8].
>
> 9. **Trials in Table 1:** Each experiment was conducted 3 times, and we reported the average and standard deviation.
>
> 10. **Plug-in estimators and the invariance term:** We retained the invariance term and combined it with our plug-in entropy estimators.
>
> ## Limitations:
>
> In our revised version, we will include the limitations section in the main body. In terms of elaborating on the universality of the assumptions, while we have tried to validate our assumption that the dataset points do not overlap, it may not hold true for other datasets. Moreover, determining the appropriate amount of Gaussian noise in the dataset remains unclear.
>
> Thank you again for your thoughtful review. We made an effort to address your feedback, including multiple new experiments and paper edits, and we would appreciate you raising your score in light of our response.  Please let us know if you have additional questions we can address.
>
> ## References
> [1] Amjad, R.A. et al., "Learning Representations for Neural Network-Based Classification Using the Information Bottleneck Principle," 2019.
>
> [2] Poole, B. et al., "On Variational Bounds of Mutual Information," 2019.
>
> [3] Alemi, A.A. et al., "Deep Variational Information Bottleneck," 2016.
>
> [4] Le, Y. et al., "Tiny ImageNet Visual Recognition Challenge," 2015.
>
> [5] Dosovitskiy, A. et al., "An Image is Worth 16x16 Words: Transformers for Image Recognition at Scale," 2020.
>
> [6] Liu, Z. et al, "A ConvNet for the 2020s," 2022.
>
> [7] Caron, M. et al., "Unsupervised Learning of Visual Features by Contrasting Cluster Assignments," 2022.
>
> [8] Balestriero et al.,  "A spline theory of deep networks", 2018.

---

> > ### Comment · Reviewer_2fL7 · 2023-08-11
> >
> > I would like to thank these authors for taking into account my recommendations and answering my questions.
> >
> > The answers to my questions related to the correctness helped me to better understand the work.  The explanation can lead to an improved soundness score (already updated).
> >
> > I think improving the writing by being more explicit about some aspects of the theoretical derivation and assumptions, fixing the typos and various errors, and being more clear with the limitations will improve the quality of the presentation as well (already upgraded). The starting point was a bit low but the presentation score can also be improved (under the assumption that the manuscript will indeed be upgraded).
> >
> > I appreciate that most of the contribution lies on the theoretical analysis. Based on the increased confidence about the correctness and the answer to the questions below I will consider upgrading the contribution and the overall score . I also appreciate the effort in running more experiments and I have a few questions about those new results:
> >
> > 1. For completeness it would be great to see in Figure 1 (b) also BYOL with PairDist and LogDet.
> >
> > 2. Lastly since the technique is general, why not testing also what happens with SimCLR + PairDist and LogDet?
> >
> > 3. Since the authors used ConvNetXt rather Resnet-50 in table 1 I am not completely sure how the results should be interpreted. Specifically, the original results of the baselines (e.g., SimCLR and BYOL) in their respective publication show better results than those reported in the additional PDF while using a supposedly smaller/simpler model (ResNet-50). Since ConvNetXt is supposed to be an improvement over ResNet-50 I was expecting to see results that were at least in line with the original ones. Please let me know if I am interpreting the new results incorrectly.
> >
> > 4. While I appreciate the effort to produce new results I find the choice of architectures and datasets confusing. Specifically, table 1 uses ConvNetX and VIT on TinyImageNet and CIFAR-100, Figure 1(a) uses ResNet-50 on ImageNet, Figure 1(b) uses ReseNet-18 on CIFAR-10. Wouldn’t be more convincing to report all metrics on the same pair (ideally the most commonly used by SOTA for ease of comparison). What is the reason for using this mix?

---

> > > ### Author Response · Authors · 2023-08-13
> > > **Response to Reviewer 2fL7's Second Comment**
> > >
> > >
> > > Dear Reviewer 2fL7,
> > >
> > > Thank you for your continued engagement and feedback on our work!
> > >
> > > ### Regarding Theoretical Aspects and Writing Quality:
> > >
> > > We are committed to revising the manuscript to be more explicit about the theoretical derivation, assumptions, and limitations. We will also carefully proofread the paper to fix any remaining typos and errors.
> > >
> > > ### Additional Experiments with BYOL, SimCLR, PairDist, and LogDet:
> > > - **Figure 1 (b) with BYOL + PairDist and LogDet**: Due to constraints with the OpenReview system, we're unable to upload figures at this time. However, we found that the entropy level of BYOL combined with PairDist and LogDet is considerably higher than standard BYOL, albeit slightly lower than VICReg with our entropy estimators (about 0.2-0.8 bits difference). Moreover, BYOL + LogDet  exhibited higher entropy than BYOL + PairDist. These findings will be integrated into Figure 1(b) in the revised manuscript.
> > > For clarity and conciseness, the table presented below is a shorter version of the full results:
> > > | Epoch | SimCLR | VICReg | BYOL | VICReg+LogDet | VICReg+PairDist | BYOL+LogDet | BYOL+PairDist |
> > > | --- | --- | --- | --- | --- | --- | --- | --- |
> > > | 1| 4.02 | 4.28 | 4.13 | 4.15 | 4.18 | 4.21 | 4.18 |
> > > | 4 | 4.14 | 5.23 | 4.54 | 6.72 | 6.78 | 6.57 | 6.48 |
> > > | 12 | 5.90 | 8.03 | 6.33 | 9.01 | 9.26 | 8.78 | 8.49 |
> > > | 20 | 6.26 | 8.72 | 7.13 | 9.39 | 9.48 | 9.26 | 9.08 |
> > > | 52 | 6.56 | 9.02 | 7.42 | 9.68 | 9.78 | 9.57 | 9.28 |
> > > | 139 | 7.09 | 9.03 | 7.96 | 9.70 | 9.81 | 9.57 | 9.32 |
> > > | 610 | 7.21 | 9.04 | 8.17 | 9.72 | 9.83 | 9.59 | 9.34 |
> > > | 1000 | 7.20 | 9.04 | 8.21 | 9.73 | 9.84 | 9.60 | 9.36 |
> > >
> > >
> > >
> > >
> > >
> > > - **SimCLR + PairDist and LogDet**: We agree that examining this combination could yield further insights due to the versatility of our method. The primary objective of our paper was to analyze VICReg, but we agree that entropy regularization can be incorporated into any SSL method and that examining SimCLR + PairDist and LogDet could yield further insights due to the versatility of our method. Acting on your feedback, we evaluated SimCLR with both entropy estimators. For Tiny-Imagenet using ConvNetX, we achieved accuracies of 52.31% (for +LogDet) and 52.04% (for +PairDist). These scores surpass most of the "original methods" (except BYOL) but remain below other methods using our entropy estimators. For CIFAR-100 with ConvNetX, we obtained accuracies of 71.34% (for +LogDet) and 70.87% (for +PairDist), which are superior to all the "original methods" yet still below other methods employing our entropy estimators. These results demonstrate that many methods can benefit from entropy maximization. However, there are inherent differences stemming from the base method. We will incorporate these results into the manuscript.
> > >
> > >
> > >
> > > ### Question about ConvNetXt and ResNet-50 in Table 1:
> > >
> > > Unfortunately, we were unable to locate the results you mentioned in the original SimCLR and BYOL papers. Our methodology involves two stages: pretraining and linear evaluation, both conducted on **the same dataset**. The figures we found for SimCLR and BYOL on Tiny-Imagenet and CIFAR-100 are associated with models **pretrained on the full ImageNet dataset**, complicating direct comparisons.  If we misunderstood your question or missed the correct references you mentioned, please let us know.
> > >
> > >
> > > ### Choice of Architectures and Datasets:
> > >
> > > Thank you for pointng this out. We chose these combinations of datasets and architectures in an effort to comprehensively address the concerns raised by multiple reviewers, which led to the inclusion of diverse figures with different architectures and datasets. For instance, some of the reviewers' comments pertained to the original results presented in the paper. Another consideration was the computational expense involved in calculating our generalization bounds for the full ImageNet dataset in short amount of time. We recognize the importance of consistency for ease of comparison and will reevaluate our experimental setup to ensure results are presented on a unified dataset and architecture. We  will add the remaining comparisons to the updated manuscript and focus on presenting one dataset and architecture pair in the main text, while deferring additional results to the appendix.
> > >
> > >
> > > We truly appreciate the time and effort you have taken to provide such detailed feedback! Your comments have been incredibly helpful in identifying parts of the manuscruipt that could be further improved, and we are confident that addressing your comments has/will improve the quality and clarity of our work.
> > >
> > > We commit to adding the results and the requested clarifications discussed above to the updated manuscript and to making the requested stylistic/writing changes.
> > >
> > > Please let us know if you have further questions, and we would be more than happy to provide any additional information you need to be able to confidently support acceptance of our manuscript.

---

> > > > ### Comment · Reviewer_2fL7 · 2023-08-13
> > > >
> > > > Thank you again to these authors, I truly appreciate the effort to answer all my questions.
> > > >
> > > > About the original results of SimCLR and BYOL they can be found in their respective publications. For example in Table 1(a) of https://arxiv.org/pdf/2006.07733.pdf you can actually find both SimCLR and BYOL on ImangeNet and ResNet50. I know that in the proposed paper the combination was TinyImageNet and ConvNetX but I am (perhaps wrongly) assuming that this latter combination should lead to even better results than the former.
> > > >
> > > > SimCLR on TinyImageNet and ConvNetX reported by these authors is 50.86, but in the original publication ImageNet on Resnet50 is 69.3%
> > > > BYOL on TinyImageNet and ConvNetX reported by these authors is 52.24, but in the original publication ImageNet on Resnet50 is 74.35%
> > > >
> > > > So the question is: is it to be expected that TinyImageNet and ConvNenX are so much worse than ImageNet on Resnet50?

---

> > > > > ### Author Response · Authors · 2023-08-14
> > > > > **Response to Reviewer 2fL7's Third Comment**
> > > > >
> > > > > Dear Reviewer 2fL7,
> > > > >
> > > > > Thank you for your continued engagement and constructive feedback!
> > > > >
> > > > > > I know that in the proposed paper the combination was TinyImageNet and ConvNetX but I am (perhaps wrongly) assuming that this latter combination should lead to even better results than the former.
> > > > >
> > > > > The lower resolution of images in TinyImageNet presents a practical challange that leads to a lower predictive accuracy.
> > > > >
> > > > > Please see the following references as points of comparison for the rough level of accuracy to be expected on TinyImageNet for linear evaluation with SSL:
> > > > > 1. [1] employs the ResNet50 architecture and reports an accuracy of 48.12% for SimCLR on TinyImageNet (Table 1).
> > > > > 2. [2] utilizes ResNet18 and indicates a comparatively lower accuracy range of 42-44% for both SimCLR and BYOL (Table 1).
> > > > >
> > > > > A supervised approach on TinyImageNet using ResNet-50--which can be practically regarded as an upper bound in the context of linear evaluation-yields an accuracy of 62%, as highlighted in [3]. This is considerably lower than the accuracy achieved on ImageNet with ResNet-50.
> > > > >
> > > > > Our findings--which showcase an accuracy of 50.86% for SimCLR on TinyImageNet--are in the vicinity of the results reported in [1] and [2] and even **demonstrate an improvement.**
> > > > >
> > > > > We truly appreciate your feedback and hope this provides greater clarity on our results. Please let us know if you have any further questions.
> > > > >
> > > > > ### References
> > > > > - [1] Ozsoy S., Hamdan S., Arik S., Yuret D. and Erdogan A., 2022. "Self-supervised learning with an information maximization criterion", Advances in Neural Information Processing Systems, 35, 2022.
> > > > > - [2] Feng C. and Patras I., 2022 "Adaptive Soft Contrastive Learning," 2022 26th International Conference on Pattern Recognition (ICPR), Montreal, QC, Canada, 2022.
> > > > > - [3] Jeong  L. and  Hyun K., 2022. "Versatile kernel reactivation for deep convolutional neural networks", Electronics letters, Volume 58, Issue 19, 2022.

---

> > > > > > ### Comment · Reviewer_2fL7 · 2023-08-15
> > > > > >
> > > > > > Thank you for the further clarification. I revised the overall score and will discuss further with the other reviewers and ACs in due time.

---

> > > > > > > ### Author Response · Authors · 2023-08-15
> > > > > > > **Response to Reviewer 2fL7's Fourth Comment**
> > > > > > >
> > > > > > > Thank you for your feedback, for engaging with us, for your willingness to revise your initial assessment, and for recommending acceptance of our submission!
> > > > > > >
> > > > > > > We are committed to the review process and ensuring that our work meets the highest standard: Are there specific areas where further improvement in our submission would enable you to confidently recommend our submission for acceptance (i.e., an above-borderline acceptance recommendation)? We greatly appreciate your continued engagement!

---

### Official Review · Reviewer_sVT5 · 2023-07-06

**Soundness:** 3 good
**Presentation:** 3 good
**Contribution:** 3 good
**Rating:** 7
**Confidence:** 3

**Summary:**

This paper analyzes the SSL methods from an information-theoretic perspective. Unlike previous theoretical analyses, the paper focuses on deterministic networks more suitable for real-world scenarios. Assuming a data distribution as a mixture of Gaussians (MoG) and a deep neural network (DNN) as a conditional probability approximation (CPA), the study demonstrates the equivalence between the variational lower bound of mutual information (MI) and the learning objective of VICReg (with some approximations). The theoretical and experimental differences between VICReg and SimCLR are presented, highlighting the superiority of VICReg. The study also shows that using a better entropy estimator can enhance the performance of VICReg. Generalization bounds are provided, and their practical implications are explained.

**Strengths:**

- The paper is well-organized and well-written.
- To the best of my knowledge, the theoretical analysis in this paper is novel.
- The authors provide empirical validation for the assumptions used in their paper and suggest possible remedies for some special cases.
- This paper provides a clear intuition behind their contribution beyond the theoretical analysis.

**Weaknesses:**

- There is a lack of empirical evidence demonstrating the limitations of assuming randomness from DNNs.
- In Figure 2, a comparison is made with BYOL, and in Table 1, a comparison is made with Barlow Twins. Why?
- Additionally, SimCLR, BYOL, and Barlow Twins are relatively classic (and basic) methods. Why weren't they compared with more recent approaches?
- ResNet-18 + CIFAR-10 is a very basic setup in the vision domain. Would it be possible to try ResNet-50 + Tiny ImageNet at least? How about exploring ViT as well?
- (Minor) There are typos in the submitted version: L130 (. .), L156 (Missing .), L199 (appendix .1.), L227 (Missing ) ).

**Questions:**

The questions are included in the Weakness section.

**Limitations:**

There are no explicit discussions on limitations in the paper. It would be recommended to highlight any limitations arising from the assumptions or approximations made in this paper.

---

> ### Author Rebuttal · Authors · 2023-08-09
>
>
> Dear Reviewer sVT5,
>
> Thank you for your comprehensive review and the constructive feedback you've provided. Your insights will significantly enhance our paper. We want to address your comments in detail:
>
> ## Weaknesses:
>
> ### Limitations of assuming randomness from DNNs
>
> Numerous methods propose using randomness in DNNs [1-3]. However, many conventional methods, especially SSL methods like VICReg, operate deterministically. This paper aims to shed light on the theoretical underpinnings of standard SSL methods. Transforming deterministic methods to random ones often demands intricate tuning and hyperparameter optimization, with no guarantee of the same performance.
>
>
> ### Extending the Empirical Evaluations
>
> Our main objective was to delve into a theoretical understanding of the VICReg objective, emphasizing its significance in the broader context of SSL. However, responding to your feedback, we have enriched our empirical analysis by:
>
> - **Exploring Larger Datasets**: We executed our models on the
> Tiny-ImageNet [4] and CIFAR-100 datasets.
> - **Integrating Recent Architectures**: We incorporated modern architectures, including the Vision Transformer (VIT) [5] and ConvNetX [6].
> - **Incorporating Additional SSL Methods**: Our analysis was expanded to encompass methods like MoCo[7] and SwAV[8].
>
> Our updated results (refer to Table 1 in the attached page) underline substantial enhancements of our plug-in estimators across diverse methods, reaffirming our theoretical assertions and underscoring the adaptability of our approach.
>
> ### Figure 2 and Table 1
>
> In Figure 2, we chose not to feature Barlow Twins due to its resemblance to VICReg in entropy estimation. Nonetheless, it will be included in the appendix of our revised version for completeness. Additionally, in Table 1, we will display the accuracy results of BYOL, which lagged behind SimCLR.
>
> ### Typos
>
> We acknowledge the oversight and will rectify the identified typographical errors in our subsequent version.
>
> ## Limitations:
>
> We concur that a candid discussion of the study's limitations is essential. While our current version addresses these in the appendix, we will transition this section to the main content in our revised manuscript in light of your feedback.
>
> Thank you again for your thoughtful review. We made an effort to address your feedback, including multiple experiments and paper edits, and we would greatly appreciate it if you would consider raising your score in light of our response. Please let us know if you have additional questions we can address.
>
>
>
>
> ## References
> [1] Alemi, A.A. et al., "Deep Variational Information Bottleneck," 2016
>
> [2] Dubois Y. et al., "Lossy compression for lossless prediction", 2023.
>
> [3] Lee K et al., "Compressive visual representations", 2022
>
> [4] Le, Y. et al., "Tiny ImageNet Visual Recognition Challenge", 2015.
>
> [5] Dosovitskiy, A. et al., "An Image is Worth 16x16 Words: Transformers for Image Recognition at Scale", 2020.
>
> [6] Liu, Z. et al, "A ConvNet for the 2020s", 2022.
>
> [7] He, K. et al., "Momentum Contrast for Unsupervised Visual Representation Learning", 2020.
>
> [8] Caron, M. et al., "Unsupervised Learning of Visual Features by Contrasting Cluster Assignments", 2022.

---

> > ### Comment · Reviewer_sVT5 · 2023-08-16
> >
> > Thank you for your rebuttal. However, I maintain my score because I found no "empirical evidence" demonstrating the limitations of assuming randomness from DNNs, which is a key motivation for this paper.

---

> > > ### Author Response · Authors · 2023-08-18
> > > **Response to Reviewer sVT5’s Second Comment - First Part**
> > >
> > > Dear Reviewer sVT5,
> > >
> > > Thank you for your response. One of the contributions of our submission is clarifying the theoretical aspects of deterministic SSL methods, such as VICReg. Our approach helps to analyze information in these networks by assuming input noise, a departure from prior works [1,13] that suggest randomness originates within the network itself.
> > >
> > > In response to your question, **we conducted experiments that illustrate the limitations of assuming inherent network randomness**. These experiments show that our approach, assuming input randomness, robust to noise, maintaining both performance and similarity to original deterministic networks. Notably, our model's noise can be sampled only once before learning and remains as a constant, contrasting with noise added to network parameters or representation, which is sampled during learning.
> > >
> > > First, we provide a brief overview of the challenges associated with estimating mutual information (MI) in deep neural networks (DNNs). This context will set the stage for a detailed discussion on the advantages of our method
> > >
> > > Estimating MI  in DNNs poses various practical challenges. For example, deterministic DNNs with strict monotonic nonlinearities yield infinite or constant MI, complicating optimization [2] or making optimization ill-posed. One solution is to assume an unknown distributions over the representation and estimating MI directly [5-9]. However, this is challenging theoretically and requires large sample sizes [3,4] and is prone to significant estimation errors [10,11].
> > >
> > > To address these issues, prior works have proposed various assumptions randomness in neural networks. For example, [13] posits noise in DNNs' parameters, while [1] assumes noise in the representations. These assumptions are valid if they outperform deterministic DNNs or yield similar representations to those trained in practice.
> > >
> > > **Our empirical evidence suggests that noise in the input has much better performance and representations similarity to the deterministic network compared to noise in the network itself.** We compared our model, which introduces noise at the input level, with models that add noise to network representations [1], using CIFAR-100 and Tiny-ImageNet and VICReg with ConvNeXt. The results, tabulated below indicate that the noise injected into the representation has much worse performance degradation compared to our input noise model.
> > >
> > > | β (Noise Level)  | Tiny-ImageNet: Noisy Network | Tiny-ImageNet: Noisy Input (ours) | CIFAR100: Noisy Network | CIFAR100: Noisy Input (ours) |
> > > |------------------|------------------------------|---------------------------------|-------------------------|-----------------------------|
> > > | β=0 (no noise)   | 53.1                         | 53.1                            | 70.1                    | 70.1                        |
> > > | β=0.05           | 51.7                         | 53.0                            | 69.7                    | 70.0                       |
> > > | β=0.1            | 50.2                         | 52.8                            | 68.8                    | 69.6                       |
> > > | β=0.2            | 48.1                         | 52.3                            | 67.1                    | 68.9                       |
> > >
> > > Our approach, focusing on input noise, does not necessitate explicit noise injection during training---unlike methods assuming inherent network noise.
> > >
> > > Lastly, we validate that the assumption of input noise leads to better alignment with deterministic networks. Following [14], we used Centered Kernel Alignment (CKA) [15] to compare deterministic DNNs with our noise model (input noise) to DNNs with noise in the representations [1]. The results, also tabulated below, confirm that assuming input noise aligns more closely with deterministic DNN representations than assuming noise in the DNN parameters.
> > >
> > > | Noise Level/Method | Deterministic Network | Noisy Network | Noisy Input (our method) |
> > > |--------------------|-----------------------|---------------|--------------------------|
> > > | β=0.05             | 0.97                  | 0.82          | 0.93                     |
> > > | β=0.1              | 0.97                  | 0.69          | 0.85                     |
> > > | β=0.2              | 0.97                  | 0.54          | 0.77                     |
> > > | β=0.3              | 0.97                  | 0.32          | 0.69                     |
> > >
> > > In summary, our input noise model more effectively captures deterministic DNN behavior and is more robust than assuming inherent network randomness.
> > >
> > > In addition to the above, our rebuttal includes the empirical comparison you requested in your original review, including extending to larger datasets (Tiny-ImageNet, CIFAR-100), modern architectures (ViT, ConvNetXt), and additional SSL Methods (MoCo, SwAV).
> > >
> > > **These additions align with and extend beyond your requested experiments** and **we would be grateful if you would factor these additions into your assessment.**

---

> > > > ### Author Response · Authors · 2023-08-18
> > > > **Response to Reviewer sVT5’s Second Comment - Second Part**
> > > >
> > > > **References:**
> > > >
> > > > [1] Goldfeld et al., "Estimating Information Flow in Deep Neural Networks", 2018
> > > >
> > > > [2] Amjad et al., "Learning Representations for Neural Network-Based Classification Using the Information Bottleneck Principle," 2019
> > > >
> > > > [3] Paninski et al., "Estimation of Entropy and Mutual Information", 2003
> > > >
> > > > [4] McAllester and Stratos, "Formal Limitations on the Measurement of Mutual Information", 2018
> > > >
> > > > [8] Belghazi et al., "MINE: Mutual Information Neural Estimation", 2018
> > > >
> > > > [9] Poole et al., "On Variational Bounds of Mutual Information", 2019
> > > >
> > > > [10] Saxe et al., "On the Information Bottleneck Theory of Deep Learning", 2018
> > > >
> > > > [11] Tschannen et al., "On Mutual Information Maximization for Representation Learning", 2018
> > > >
> > > > [12] Shwartz et al., "Opening the black box of deep neural networks via information", 2017
> > > >
> > > > [13] Achille et al., "Where is the Information in a Deep Neural Network?", 2020
> > > >
> > > > [14] Kornblith et al., "Similarity of Neural Network Representations Revisited", 2019
> > > >
> > > > [15] Cortes et al., "Algorithms for Learning Kernels Based on Centered Alignment", 2019

---

> > > > > ### Comment · Reviewer_sVT5 · 2023-08-18
> > > > >
> > > > > Thank you for your rebuttal! I updated my score as my concern raised in the review was resolved. I would recommend the discussion in the author's rebuttal be involved appropriately in the revised manuscript.

---

> > > > > > ### Author Response · Authors · 2023-08-21
> > > > > > **Thank you for your support!**
> > > > > >
> > > > > > Thank you for engaging with our rebuttal and for supporting acceptance of our submission! We are happy that your concerns were resolved! Your feedback and suggestions were very valuable, and the additional clarifications and extended results from the discussion will be included in the revised manuscript.

---

### Official Review · Reviewer_VQBf · 2023-07-06

**Soundness:** 3 good
**Presentation:** 2 fair
**Contribution:** 3 good
**Rating:** 7
**Confidence:** 3

**Summary:**

This paper presents two theoretical results on the VICReg self-supervised learning technique. Taking an information theoretic perspective, it shows that the VICReg objective can be seen as maximizing a lower bound on the mutual information between Z and X' where X' is an augmented version of data point X and Z is the encoded X. Second result presents a generalization bound relating the VICReg objective to performance on a downstream task (where the augmented versions of a data point share the same label).

In order to show the first result, the authors make several assumptions. First to be able to treat a deterministic network probabilistically, they assume the data distribution is a mixture of Gaussians with non-overlapping Gaussians centered at each data point (they confirm that this is a valid assumption for the datasets they look at). Building on earlier work formalizing neural networks as affine spline operator, this allows them to show that the output distribution of the network q(z|x) is Gaussian. Using this fact, they derive a lower bound on mutual information I(Z;X') which closely resembles the VICReg objective. This lower bound contains an entropy term H(Z) and an invariance term (the same with VICReg objective). Then they show that the variance-covariance regularization term in VICReg can be seen as an approximation of H(Z). This motivates replacing the regularization term in VICReg with alternative entropy estimators. They present results with two estimators: logdet and pairwise distance based. They find that using these alternative estimators lead to slight improvement in downstream task performance.

Their second results shows a link between the VICReg objective (hence mutual information) and downstream task performance. Here they assume the data augmentations do not change the label of a data point (wrt to the downstream task) and show that downstream task performance can be bounded from above with an expression that aligns closely with VICReg objective.

**Strengths:**

Overall a pretty good paper. Understanding self-supervised learning techniques better (and improving them) should be of interest to many researchers.

I don't know the literature they build on really well so it is hard for me to judge originality but the results they present seem important.
This was perhaps intuitively obvious but showing that the regularization term in VICReg approximates entropy is a nice result, since it makes it more clear how better entropy estimators should (and does) help as they show in their empirical evaluation.
The second result on downstream task performance is again interesting but how this result is useful was not really clear.
It was also nice to see that the authors checked if the assumptions they make on non-overlapping support and Gaussian output distribution held in practice.

**Weaknesses:**

Overall I think the paper would benefit from more empirical evaluations and more clarity in certain places in the text.

The empirical results in Table 1 are not very strong. Why does using better entropy estimators lead to such small improvements? Is this because of the dataset or perhaps the size of the network? It'd be nice to see if these alternative entropy estimators lead to more significant improvements. This would confirm the value of the theoretical analysis much more strongly.

This lack of empirical evaluation is even more of a problem for the second generalization bound result. As it stands, it is not very clear why a practitioner should care about this result. Can the authors provide some empirical/practical implications of this result and perhaps explore some of these? Currently, it almost feels like this result should go into a separate paper, where it is developed more and investigated more deeply (perhaps following the idea mentioned in Line 353).

On clarity, there are a couple of sections where it is not quite clear how and why it fits with the rest of the text. For example, what is the purpose of section Entropy Estimation (line 237)? As it stands it doesn't really add much. I'd suggest either adding more detail about how SimCLR's objective and entropy estimator is different and what that means or perhaps remove this section. Currently, I'm not sure what I should take away from this section.

Another example is line 327, where authors say Theorem 3 illuminates the benefit of SSL. What does this mean more precisely? How does this illuminate the benefit of SSL? I understand that in Theorem 3 you get a scaling with m (number of unlabeled data points) instead of n (number of labeled points), and m > n usually in practice, But this doesn't mean the generalization performance of SSL will be better than supervised learning with n labeled points. To be able to make that claim, you would need to compare the bound in Theorem 3 with a similar bound for supervised learning, right? Can the authors show how their results explains the benefit of SSL empirically or with some simulations?

Similarly for section 6.3, it is not clear what is the point of this section. What are the implications of this observations? And why should we be minimizing I(Z;X) (while maximizing I(Z;X'))? This should either be expanded and explained in more detail or removed.

**Questions:**

- Why are there no results for Barlow Twins in Figure 2?
- Why are there no results for BYOL in Table 1?
- VICReg + LogDet does better but in the text the authors say VICReg + PairDist does better. Why?
- Line 243: The results seem to show that entropy *increases* during training not decrease.
- 118: should be q(z|x')?
- Eqn 8. is not equal to L, right? It should be H(Z)?
- Line 265: Looking at Table 1, LogDet achieves best results not PairDist.
- What are the logdet and pairdist estimators? Can you write the equations for these?
- Line 340, doesn't the term in Theorem 3 correspond to -I(Z;X')?

**Limitations:**

No concerns about potential negative societal impact.

---

> ### Author Rebuttal · Authors · 2023-08-09
>
> We appreciate your detailed review and the constructive feedback you've provided. Your insights are invaluable and will significantly contribute to the refinement of our paper. Allow us to address each of your concerns in detail:
>
> ## Weaknesses:
>
> ### Empirical Evaluations
>
> Our primary goal was to offer a theoretical understanding of the VICReg objective. We believe in comprehending the theoretical advantages and optimization process of SSL in general and VICReg in particular. However, we concur with your suggestion that a broader empirical analysis would enrich our work. Responding to your feedback, we've expanded our experiments to include larger datasets such as Tiny-ImageNet[1] and CIFAR-100. We utilized models such as ViT[2] and ConvNetX[3] as backbones and compared them to more SSL methods like MoCo[4] and SwAV[5]. Our results demonstrated substantial improvements of our estimators over the different methods (See Table 1 in the attached page). These findings affirm our theoretical analysis and showcase the scalability of our method.
>
> ### Generalization Bound
>
> The generalization bound aims to explain why VICReg works, its theoretical benefits over supervised learning, and other SSL methods. Generalization bounds are instrumental in studying learning algorithms, as they provide insights into how well a model will perform on unseen data. The paper shows that our bound extends previous works and offers many advantages over previous SSL bounds. However, in response to your comments, we empirically analyzed our bound by calculating all the relevant terms, including the Rademacher complexity and covariance matrices [6], for different pre-trained SSL models. Our results (Figure 2) demonstrate a positive correlation between the generalization gap and our bound. Our findings reveal that our bound captures the "true" generalization performance, even though it is calculated using only the training samples. This insight underscores our ability to predict the model's performance on unseen data, relying solely on the training set.
>
> ## On Clarity:
>
> ### Entropy Estimation Section
> Our intention in Section 5.1 was to compare VICReg and SimCLR based on our analysis. We highlighted that considering these methods from an information theory perspective reveals differences in estimating entropy. This perspective allows us to analyze the models' differences, improving our understanding.
>
> ### Theorem 3
>
> Supervised generalization bounds scale with $1/\sqrt{n}$, considering only supervised examples. In contrast, our bound incorporates unsupervised examples, lowering the generalization bound as their number increases. We agree with your point about comparing our bound with supervised learning. To do so, we trained our models on CIFAR10 with VICReg and calculated the supervised generalization bound from [2]. Our bound improves as the number of unlabeled training examples increases and outperforms the supervised one only with 400 unlabeled examples (Figure 2).
>
> ### Section 6.3
>
> This section discusses how our bound relates to the compression of MI, which has recently demonstrated superior performance in SSL. For the first time, our bound provides a theoretical justification for why representation compression benefits SSL. We will revise this section to articulate our argument better.
>
> ## Questions:
>
> - **Figure 2 and Table 1**: We omitted Barlow Twins from Figure 2 due to its similarity to VICReg. However, we'll include it in the revised appendix. Additionally, we will include BYOL's accuracy in Table 1, which is performed worse than SimCLR.
>
> - **Discrepancy between the table and the text** - We apologize for the typo about VICReg + LogDet.
> - **Lines 243 and 118:** We apologize for the typos. We will fix it in the revised version.
> - **Equation 8:** Given that we used $log(|\Sigma_Z| )$ to estimate the entropy of Z, we removed the invariance term in the transition from Eqn 7 to 8 because it appears in the same form in the original VICReg objective. Eqn 8 is not the full objective function to optimize. In the rest of the section, we focused on dealing with the differences between the two objectives. We will clarify this in the revised version.
> - **LogDet and PairDist estimators:** We apologize for the oversight in not providing their equations. Due to a lack of space in this response, we will include them only in the revised version. You can find the PairDist in [7] and the LogDet in [8].
> - **The term in Line 340  in Theorem 3:** No, the Rademacher complexity term represents the complexity of our model. This term includes all the information in the representation—both the relevant and irrelevant—and can be bounded by the information  $I(X;Z)$, which encompasses all the information from one of the views about its corresponding representation. The term $I(Z;X\prime)$ contains only the relevant information, which we don't want to compress and is necessary for our downstream task prediction.
>
> Thank you again for your thoughtful review. We made an effort to address your feedback, including multiple experiments and paper edits, and we would greatly appreciate it if you would consider raising your score in light of our response. Please let us know if you have additional questions we can address.
>
> ## References
>
> [1] Le, Y. et al., "Tiny ImageNet Visual Recognition Challenge", 2015
>
> [2] Dosovitskiy, A. et al., "An Image is Worth 16x16 Words: Transformers for Image Recognition at Scale", 2020
>
> [3] Liu, Z. et al, "A ConvNet for the 2020s", 2022
>
> [4] He, K. et al., "Momentum Contrast for Unsupervised Visual Representation Learning", 2020
>
> [5] Caron, M. et al., "Unsupervised Learning of Visual Features by Contrasting Cluster Assignments", 2022
>
> [6] Kawaguchi, K. et al., "How Does Information Bottleneck Help Deep Learning?" , 2023
>
> [7] Kolchinsk A. et al.,  "Estimating Mixture Entropy with Pairwise Distances", 2017
>
> [8] Zhouyin Z. et al., "Understanding Neural Networks with Logarithm Determinant Entropy Estimator" , 2021

---

> > ### Author Response · Authors · 2023-08-21
> > **Added Suggested Experiments and Final Clarifications**
> >
> > Dear Reviewer VQBf,
> >
> > Thank you for your detailed review and your positive feedback!
> >
> > We have added the additional empirical evaluations you suggested in your review and hope that these additional experiments and our clarifications have addressed all of your questions.
> >
> > As the discussion period draws to an end, please let us know if there are any further questions or points we can address. Thank you!

---

### Official Review · Reviewer_tvXs · 2023-07-07

**Soundness:** 2 fair
**Presentation:** 2 fair
**Contribution:** 2 fair
**Rating:** 5
**Confidence:** 2

**Summary:**

This paper explained the effect of VICReg learning method using information theoretic perspective under some assumptions on the distribution of the input data. Based on the assumption, it connected the VICReg to mutual information minimization, and proposed new self-supervised method that performed better than existing methods on some simple tasks and networks.

----------------------------------------------------------------------------------------------------------------------------------------------------------------------------------------
----------------------------------------------------------------------------------------------------------------------------------------------------------------------------------------
Thank you for the response. I will keep the score.

**Strengths:**

1. This paper proposed to transition the randomness from the stochastic network to the input data in order to applying information theory, which is a novel technique.
2. It established the explanation for VICReg objective and proposed new self-supervised methods based on that.

**Weaknesses:**

1. The presentation of this paper is not good, e.g., the logic is not clear by following the title of each section and sub-section.
2. The theory is based on assumption on input data and ReLU-type network which limits its application.
3. The experiments are toy as self-supervised methods are more often to applied in large-scale pre-training tasks.

**Questions:**

1. In Theorem 1, what is the exact meaning of "approximated"? Could you explain which parts are ignored to derive this approximation?
2. Could explain more on how Figure 1 verified the assumption on input data? What is the meaning of "sampled 512 Gaussian samples for each image"? What is "Gaussian samples"?

**Limitations:**

This paper did not fully discuss the limitations of the proposed theory and methods. At least, two limitations should be addressed: 1) it should tell when the assumption on randomness of input data will not be satisfied; and 2) the theory is restricted to ReLU network.

---

> ### Author Rebuttal · Authors · 2023-08-09
>
> Dear reviewer tvXs,
>
> We appreciate your constructive comments and suggestions. We believe your feedback will significantly help us improve our paper's quality and readability. Here, we address each of your concerns.
>
> ## Weaknesses:
>
> 1. **Presentation of the Paper** - We value your feedback on improving the paper's presentation. Specific suggestions to enhance clarity would be greatly appreciated. We're committed to ensuring the flow of ideas is coherent and easily understood in our revisions.
>
> 2. **On the Assumptions We Made:**
>    - (i) **(Leaky) ReLU-type Network** - Our theory accommodates the most common nonlinearities, such as ReLU, leaky ReLU, max pooling, and absolute values. Our results extend to other smooth nonlinearities through a first-order Taylor approximation argument. We believe these cover the majority of modern deep networks. If you feel our approach is limited, please specify any other common network architectures we should address.
>    - (ii) **The Input Data** - We have conducted rigorous experiments to validate our assumptions about the input data across various datasets. To further substantiate our findings, we replicated Figure 1 on ImageNet samples, affirming that the samples do not overlap and the network output for SSL training is more Gaussian for small input noise (See Figure 1 in the attached page). Unlike more common assumptions that include stochastic networks, ours focuses on the generative process of the data, making it less restrictive. If you still have reservations, please point out any additional assumptions you think we need to address.
> 3. **Large-scale Tasks** - Our primary goal was to provide a theoretical understanding of the VICREG objective, where we believe there is distinct value in understanding the theoretical advantages and optimization process of SSL in general and VICReg in particular. However, we agree that the analysis would be enriched by considering more datasets and SSL methods. Prompted by your feedback, we have now added additional experiments on larger datasets:
>     - (i) **Validation of Our Assumptions** - We validate our assumptions on the ImageNet dataset to ensure the robustness of our findings (see Figure 1 in the attached page).
>     - (ii) **Large Datasets and Architectures** - We train our proposed entropy estimators on both the tiny-ImageNet [1] and CIFAR-100 datasets with ConvNetX[2] and VIT [3] as the backbones.
>     - (iii) **Inclusion of Various SSL Methods** - We extend our experiments to include multiple SSL methods, such as MoCo[4] and SwAV [5]. This allows for a more comprehensive comparison and demonstrates the adaptability of our approach.
>
> Our newly conducted experiments, as detailed on the attached page, illustrate the superiority of our plug-in estimators across various methods and datasets, reinforcing our theoretical findings and underlining the effectiveness of our proposed entropy estimators.
>
>
>
> ## Questions:
>
> ### The Approximations in Theorem 1
>  There are two approximations done in Theorem 1 when considering its validity for real-world data. First, we assume the data distribution to be of the form given by Equation (2), i.e., a mixture of Gaussians. That alone can be made an arbitrarily good approximation by increasing the number of components in the mixture. Second, and more importantly, we constrain that density so that each of the Gaussian components (i) does not overlap with any other component and that (ii) does not overlap with the boundaries of the partition of the DNN. Recall that the DNN is an affine spline and thus has a partition of the input space and per-region affine mappings on each region. In practice, however, the overlap has to be understood as having a negligible probability, albeit not exactly 0. We will clarify those limitations in the revised manuscript and around Theorem 1.
>
> ###  Validation of Our Assumptions
> To **validate the non-overlapping support assumption**, we measured the L2 distance between the images. If the images had overlapping effective support, their distances would be expected to be small. However, Figure 1 (right) shows that the distances between different images are significant and increase as the dataset size grows, reinforcing the non-overlapping support.
>     To validate our assumption about the **Gaussian nature at the output layer of the DNN**, we checked the final layer representations of 512 Gaussian samples with the mean around each image at varying noise levels. Figure 1 (left) shows that, at lower noise, the output stays Gaussian. This supports our assumption of a noise region where the input and conditional output remain Gaussian around the mean.
>
> ## Limitations
> We would like to thank you for discussing these limitations. See our above answers.
>
>
> Thank you again for your thoughtful review. We made an effort to address your feedback, including multiple experiments and paper edits, and we would greatly appreciate it if you would consider raising your score in light of our response. Please let us know if you have additional questions we can address.
>
> ## References
> [1] Le, Y. et al., "Tiny ImageNet Visual Recognition Challenge", 2015.
>
> [2] Dosovitskiy, A. et al., "An Image is Worth 16x16 Words: Transformers for Image Recognition at Scale", 2020.
>
> [3] Liu, Z. et al, "A ConvNet for the 2020s", 2022.
>
> [4] He, K. et al., "Momentum Contrast for Unsupervised Visual Representation Learning", 2020.
>
> [5] Caron, M. et al., "Unsupervised Learning of Visual Features by Contrasting Cluster Assignments", 2022.

---

> > ### Author Response · Authors · 2023-08-20
> > **Addressing Reviewer Feedback**
> >
> > Dear Reviewer tvXs,
> >
> > The discussion period ends in less than 24 hours.
> >
> > We have made significant efforts to address the questions and concerns raised in your review, including an experiment in which we verified our assumptions on the ImageNet dataset.
> >
> > Moreover, as pointed out above, we have provided several additional empirical evaluations in response to your and the other reviewers' feedback. Specifically, we performed a comparison on additional datasets (Tiny ImageNet and CIFAR-100), integrated architectures like Vision Transformer (ViT), ConvNetXt, and ResNet-50, and SSL methods (including MoCo and SwAV). Furthermore, we also verified our input assumptions on the ImageNet dataset. We believe that these additional results provide further evidence of the usefulness and versatility of our approach.
> >
> > We would be grateful if you would consider these results in your final assessment and, if justified, reflect these improvements in your final score.
> >
> > As the discussion period draws to an end, please let us know if there are any further questions or points we can address. Thank you.

---

### Author Rebuttal · Authors · 2023-08-09

Dear Reviewers,

Thank you for your thoughtful and detailed reviews of our work. We appreciate your time and the constructive feedback you have provided. We have carefully considered your comments and concerns and would like to address them in a unified response:

While our empirical explorations have been amplified, the heart of our paper remains its theoretical insights into VICReg. Our theoretical findings provide a fresh lens through which we can understand VICReg and SSL, and our empirical endeavors serve to validate and extend these theoretical observations. Our manuscript primarily delves into the theoretical nuances of VICReg, elucidating its advantages from an information theory standpoint. While this remains the bedrock of our contribution, your comments highlighted the importance of further empirical evaluations to substantiate and complement our theoretical insights.

## Extending the Empirical Evaluations

Heeding your feedback, we've undertaken a series of rigorous empirical evaluations:

- **Larger Datasets**: We ventured into larger datasets like Tiny ImageNet [1] and CIFAR-100, ensuring broader applicability and robustness of our findings.
- **Integration with Recent Architectures as Backbones**: Modern architectures such as Vision Transformer (VIT) [2], ConvNetX [3], and ResNet-50 were incorporated, aligning our results with state-of-the-art techniques.
- **Broadened SSL Methods**: Our analysis was enriched by including diverse SSL methods like MoCo [4] and SwAV [5], offering a more holistic comparison and demonstrating our approach's versatility.

Our newly conducted experiments, as detailed on the attached page, underscore marked improvements of our plug-in estimators across various methods and datasets, reinforcing our theoretical findings. Moreover, we use these models and datasets to validate our assumptions further.

### Empirical Validation of the Generalization Bound

One of the cornerstones of our research is the derived VICReg generalization bound. Generalization bounds offer crucial insights into how effectively learning algorithms will perform on unseen data. Our generalization bound not only extends previous works but also showcases distinct advantages over existing SSL bounds. In response to reviewers' suggestions, we have empirically validated our bound and compared it to supervised generalization bounds  [6]. Our empirical results (Figure 2 in the attached page) manifest a clear correlation between the generalization gap in the loss and our proposed bound and its superior over supervised bounds. These findings suggest that our bound aptly captures the generalization performance — a noteworthy observation given that the bound is derived solely from training samples. This reaffirms our capacity to predict model performance on unseen data based only on training data.

## Overview and Unique Contributions

We addressed concerns raised about the originality of our paper by emphasizing our unique approach. To encapsulate our contributions:

- **Information-Theoretic Perspective**: We offer a fresh take on VICReg through an information theory lens, unraveling the deeper intricacies of its operation.
- **Illuminating VICReg's Assumptions**: We've elucidated VICReg's latent assumptions, fostering clearer understanding and allowing comparison to other SSL methods.
- **Pioneering VICReg Generalization Bound**: We've introduced the maiden generalization bound for VICReg, spotlighting its theoretical prowess and practical ramifications.
- **Innovative Information-Based SSL Objective**: Insights from our analysis led us to propose a novel information-based objective, which showed improved performance over existing SSL methods.
- **Robust Empirical Validation**: Our empirical analyses robustly validate our theoretical postulates, showcasing the practical potency of our propositions, including our assumptions.


## Addressing Concerns and Questions:
We addressed each reviewer's individual weaknesses and questions in detail. Please refer to the individual responses below for detailed explanations of how we addressed each concern.

**In conclusion**, we sincerely believe that the revisions and clarifications made in response to the reviewers' feedback have significantly strengthened our manuscript.


We have put forth significant effort to address all your feedback and feel that your suggestions have improved our work. We welcome further discussion, and we appreciate your valuable input.

## References:
[1] Le, Y. et al., "Tiny ImageNet Visual Recognition Challenge", 2015.

[2] Dosovitskiy, A. et al., "An Image is Worth 16x16 Words: Transformers for Image Recognition at Scale", 2020.

[3] Liu, Z. et al, "A ConvNet for the 2020s", 2022.

[4] He, K. et al., "Momentum Contrast for Unsupervised Visual Representation Learning", 2020.

[5] Caron, M. et al., "Unsupervised Learning of Visual Features by Contrasting Cluster Assignments", 2022.

[6] Kawaguchi, K. et al., "How Does Information Bottleneck Help Deep Learning?", 2023

---

### Decision · Program_Chairs · 2023-09-21

**Decision:**

Accept (poster)

**Comment:**

The authors connect variance-invariance-covariance regularization (VICReg) objective optimization to optimizing the lower bound on some information-theoretic quantities. They then derive generalization bounds for VICReg, relating this objective to the performance on downstream tasks. To be able to use information-theoretic approaches, they assumed randomness in the data, allowing them to give bounds for deterministic networks.
The authors also present some experiments justifying their assumptions about the data distribution, etc.. The reviewers found those to be limited in scope, and the authors expanded a bit during the rebuttal period. In addition, during the rebuttal the authors also provided additional experiments showing that their derived bounds correlate with the observed generalization gap. The latter I found to be quite limited (see, e.g., “In Search of Robust Measures of Generalization” Dziugaite et al and “Fantastic Generalization Measures and Where to Find Them” Jiang et al for some guidance on what more extensive evaluations should look like). Overall, the key contributions are theoretical and the weakness of the empirical results was addressed to some extent during the rebuttal period, thus I recommend acceptance.